# Latent Space Factorization in LoRA

**Shashi Kumar**[1,2]   **Yacouba Kaloga**[1]   **John Mitros**[1]   **Petr Motlicek**[1,3]   **Ina Kodrasi**[1]

[1]Idiap Research Institute, Switzerland
[2]EPFL, Switzerland   [3]BUT, Czech Republic
{shashi.kumar, yacouba.kaloga, petr.motlicek, ina.kodrasi}@idiap.ch
john.mitross@gmail.com

## Abstract

Low-rank adaptation (LoRA) is a widely used method for parameter-efficient finetuning. However, existing LoRA variants lack mechanisms to explicitly disambiguate task-relevant information within the learned low-rank subspace, potentially limiting downstream performance. We propose Factorized Variational Autoencoder LoRA (FVAE-LoRA), which leverages a VAE to learn two distinct latent spaces. Our novel Evidence Lower Bound formulation explicitly promotes factorization between the latent spaces, dedicating one latent space to task-salient features and the other to residual information. Extensive experiments on text, audio, and image tasks demonstrate that FVAE-LoRA consistently outperforms standard LoRA. Moreover, spurious correlation evaluations confirm that FVAE-LoRA better isolates task-relevant signals, leading to improved robustness under distribution shifts. Our code is publicly available at: `https://github.com/idiap/FVAE-LoRA`

## 1   Introduction

Foundation models have become ubiquitous across modalities such as vision [1, 2, 3, 4], audio [5, 6], and text [7, 8]. Recent state-of-the-art results are predominantly achieved by fine-tuning these large pre-trained models. Among various parameter-efficient fine-tuning (PEFT) strategies [9, 10, 11, 12], *Low-Rank Adaptation (LoRA)* [12] has emerged as a particularly efficient approach. In LoRA, the original weight matrices $\mathbf{W} \in \mathbb{R}^{k \times d}$ are kept frozen, and trainable low-rank matrices $\mathbf{A} \in \mathbb{R}^{r \times d}$ and $\mathbf{B} \in \mathbb{R}^{k \times r}$ are introduced, with $r \ll min(d, k)$, such that the adapted weights become $\mathbf{W} + \mathbf{BA}$. This technique significantly reduces memory and computational requirements, while achieving performance comparable to full fine-tuning [12, 13].

Despite the remarkable performance shown by LoRA across a plethora of downstream tasks and modalities, we identify a potential limitation: the standard LoRA update mechanism lacks an explicit way to ensure that the learned low-rank subspace $Im(\mathbf{A})$ primarily captures task-salient information. The projection $\mathbf{Ax}$ (where $\mathbf{x}$ is the input activation) is learned implicitly through gradient descent on the task objective. While effective, this process does not inherently guarantee that $\mathbf{A}$ isolates features crucial for the downstream task from potentially irrelevant or even detrimental information retained from pre-training. This lack of explicit control over the content of the low-rank update is pertinent. While the hypothesis that fine-tuning primarily involves low-rank updates provides a strong theoretical underpinning for LoRA [14], empirical evidence suggests nuances. Recent studies have shown that standard LoRA can still underperform full fine-tuning in certain scenarios [12, 15]. This suggests that simply constraining the update to be low-rank might not be sufficient; the task-relevant signal encoded within that low-rank adaptation is critical for achieving optimal downstream performance. Existing LoRA variants do not offer a principled mechanism to explicitly disentangle and prioritize task-relevant information within the learned update.

To address this limitation and enable explicit control over the information captured within the low-rank update, we propose **Factorized Variational Autoencoder LoRA (FVAE-LoRA)**. Our approach

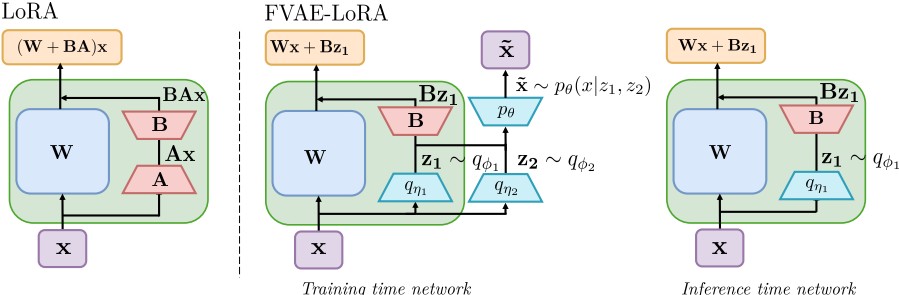

Figure 1: Comparison between LoRA and the proposed FVAE-LoRA. During training, FVAE-LoRA factorizes the latent space into two components, $z_1$ and $z_2$, where only the task-salient latent factor $z_1$ is propagated downstream. At inference, only the encoder corresponding to $z_1$ is required.

integrates a Variational Autoencoder (VAE) framework directly into the LoRA parameterization. Crucially, unlike standard VAEs, FVAE learns **two distinct latent spaces**, denoted by $z_1$ and $z_2$ (see Figure 1). We introduce a novel Evidence Lower Bound (ELBO) formulation specifically designed to promote **factorization** between these two spaces during training. This objective encourages the model to encode task-salient information, critical for downstream performance, primarily within the first latent space $z_1$, while relegating residual information necessary for accurate reconstruction (as required by the FVAE objective) to the second latent space $z_2$. During the forward pass for the downstream task, only samples drawn from the task-salient latent space $z_1$ are utilized to generate the effective low-rank adaptation matrix $A$. This mechanism allows FVAE-LoRA to explicitly select and leverage the most relevant learned features for the target task, while isolating potentially less useful or confounding variations within $z_2$.

Our main contributions can be summarized as follows:

- **A Novel PEFT Method (FVAE-LoRA):** We propose FVAE-LoRA, integrating a VAE with factorized latent spaces ($z_1$, $z_2$) into the LoRA framework to explicitly disentangle task-salient information ($z_1$) from residual information ($z_2$).

- **Factorizing ELBO Formulation:** We introduce a novel ELBO objective specifically designed to enforce this factorization between the two latent spaces during training.

- **Strong Empirical Performance:** We demonstrate through extensive experiments on diverse image, text, and audio benchmarks that FVAE-LoRA consistently outperforms LoRA.

- **Empirical Validation of Robustness:** We empirically validate, using targeted spurious-correlation experiments, that the task-salient latent space $z_1$ indeed captures task-critical information, leading to a robust performance even on challenging examples designed to mislead standard LoRA.

## 2 Related Work

We position FVAE-LoRA relative to PEFT methods, specifically LoRA variants, and techniques for latent space factorization in VAEs.

**PEFT** methods adapt large pre-trained models with minimal trainable parameters, overcoming the costs of full fine-tuning. Common approaches include inserting Adapter modules [9], optimizing continuous prompts or prefixes [10, 16], or tuning only bias terms [17]. LoRA [12] is a prominent PEFT technique that injects trainable low-rank matrices ($\Delta W = BA$, rank $r \ll min(d, k)$) into the model layers. Its efficiency and performance have led to wide adoption [14]. FVAE-LoRA builds upon the LoRA framework, aiming to enhance its effectiveness.

**LoRA Variations.** Several methods have extended LoRA. AdaLoRA [18] adaptively allocates rank budgets. DoRA [15] decouples weight magnitude and direction, applying LoRA to the latter. LoRA+ [19] adjusts LoRA's optimization by using different learning rates for its two low-rank matrices. PiSSA [20] focuses on initializing the LoRA matrices in a way that better approximates full fine-tuning updates, typically setting $A = 0$ and initializing $B$ based on principal components of the

gradient. RS-LoRA [21] aims to stabilize training and prevent rank collapse or excessive growth by incorporating regularization related to the singular values of the update matrix. Other contributions combine LoRA with quantization for further compression [22, 23]. These variants primarily modify the update's structure, optimization, or compression. Our work differs by focusing on the *semantic content* of the update. FVAE-LoRA introduces a VAE with factorized latent spaces $(\mathbf{z}_1, \mathbf{z}_2)$ and a novel ELBO to explicitly separate task-salient information $(\mathbf{z}_1)$ used for the update from residual information $(\mathbf{z}_2)$, thereby controlling the information encoded in the low-rank adaptation.

**Factorization and Disentanglement in VAEs.** VAEs [24] learn latent representations by maximizing the ELBO. Significant research focuses on learning *disentangled* representations, where latent dimensions capture independent factors of data variation [25]. Techniques often modify the ELBO, such as $\beta$-VAE [26], FHVAE [27], FactorVAE [28], TCVAE [29], and DIP-VAE [30], or employ annealing strategies [31]. While we use a VAE and aim for factorization, our goal is distinct. We do not seek to disentangle underlying data factors. Instead, FVAE-LoRA employs a novel ELBO to factorize the latent space specifically for PEFT: separating information crucial for the *downstream task* $(\mathbf{z}_1)$ from other sources of variation needed for reconstruction $(\mathbf{z}_2)$. This task-conditional factorization within the LoRA update mechanism represents the core novelty of our VAE application.

# 3 Method

In this section, we present our low-rank adaptation approach based on a VAE. We begin by briefly reviewing the standard VAE framework, and then introduce our proposed Factorized VAE (FVAE). We highlight key properties of this formulation and conclude by describing how it enables the construction of an efficient low-rank adaptation model.

## 3.1 Variational Autoencoder Objective

Consider a dataset $X \in \mathbb{R}^{n \times d}$, where observations are generated according to the process $\mathbf{x} \sim p_\theta(\mathbf{x}|\mathbf{z})$, with $\mathbf{z}$ a latent variable. The goal is to recover a latent representation $\mathbf{z}$ that explains the observations $\mathbf{x}$, which requires computing the posterior $p_\theta(\mathbf{z}|\mathbf{x})$. As this posterior is generally intractable, we introduce an approximate distribution $q_\phi(\mathbf{z}|\mathbf{x})$. It can be shown (see Appendix A.1) that the log-likelihood admits a lower bound, known as the *Evidence Lower Bound (ELBO)*:

$$\mathcal{L}^{\text{VAE}}_{\theta,\phi}(\mathbf{x}) = \mathbb{E}_{\mathbf{z} \sim q_\phi(\mathbf{z}|\mathbf{x})} [\log p_\theta(\mathbf{x}|\mathbf{z})] - D_{\text{KL}}(q_\phi(\mathbf{z}|\mathbf{x}) \,\|\, p(\mathbf{z})). \tag{1}$$

The ELBO is used as a tractable surrogate objective for maximizing $\log p_\theta(\mathbf{x})$. The first term encourages accurate reconstruction, while the second term regularizes the latent space by aligning the approximate posterior with the prior $p(\mathbf{z})$.

## 3.2 Factorized Variational Autoencoder Objective

The primary goal of FVAE is to factorize the information contained in $\mathbf{x}$ such that it is represented by two independent latent variables $\mathbf{z}_1$ and $\mathbf{z}_2$. This factorization is learned jointly with a downstream task loss applied specifically to $\mathbf{z}_1$, which guides the decomposition by encouraging $\mathbf{z}_1$ to capture task-relevant information, while $\mathbf{z}_2$ absorbs the remaining variability. Classical VAEs serve as a natural starting point to build such a model.

### 3.2.1 Preliminaries

The derivation of the classical VAE can be extended by assuming that $\mathbf{x}$ arises from a generative process involving two independent latent variables $\mathbf{z}_1$ and $\mathbf{z}_2$, with $p(\mathbf{z}_1, \mathbf{z}_2) = p_1(\mathbf{z}_1) p_2(\mathbf{z}_2)$. Additionally, we assume that the approximate posterior factorizes as $q_\phi(\mathbf{z}_1, \mathbf{z}_2|\mathbf{x}) = q_{\phi_1}(\mathbf{z}_1|\mathbf{x}) q_{\phi_2}(\mathbf{z}_2|\mathbf{x})$. Considering $\mathbf{z}_1 \sim q_{\phi_1}(\mathbf{z}_1|\mathbf{x})$ and $\mathbf{z}_2 \sim q_{\phi_2}(\mathbf{z}_2|\mathbf{x})$, the ELBO is given by

$$\mathcal{L}^{\text{VAE2LAT}}_{\theta,\phi}(\mathbf{x}) = \mathbb{E}_{\mathbf{z}_1, \mathbf{z}_2} [\log p_\theta(\mathbf{x}|\mathbf{z}_1, \mathbf{z}_2)] - D_{\text{KL}}(q_{\phi_1}(\mathbf{z}_1|\mathbf{x}) \,\|\, p_1(\mathbf{z}_1)) - D_{\text{KL}}(q_{\phi_2}(\mathbf{z}_2|\mathbf{x}) \,\|\, p_2(\mathbf{z}_2)). \tag{2}$$

This objective mirrors the standard VAE but extends it to the multi-latent setting. However, even though both the prior and the variational posterior are factorized, the model is not explicitly encouraged to selectively assign information to $\mathbf{z}_1$ or $\mathbf{z}_2$.

### 3.2.2 FVAE

To promote factorization, we introduce a regularization term that penalizes the similarity between $q_{\phi_2}(\mathbf{z}_2|\mathbf{x})$ and the uninformative prior $p_1(\mathbf{z}_1)$. Since $q_{\phi_1}(\mathbf{z}_1|\mathbf{x})$ is encouraged to align with $p_1$, this term prevents $q_{\phi_2}$ from encoding information in the same region of the latent space. Incorporating this into Equation (2), we obtain the objective

$$\max_{\theta,\phi_1,\phi_2} \quad \mathcal{L}_{\theta,\phi}^{\text{VAE2LAT}}(\mathbf{x}) + \mathbb{E}_{\mathbf{z}_1,\mathbf{z}_2}\left[\log \frac{q_{\phi_2}(\mathbf{z}_2|\mathbf{x})}{p_1(\mathbf{z}_1)}\right]. \tag{3}$$

To clarify the role of each component in Equation (3) and relate them to familiar VAE structures, we reorganize the objective using straightforward algebraic manipulations. In doing so, we isolate the standard reconstruction and KL divergence terms, and separate out the new cross-prior regularizer. Introducing scalar constants $\alpha$, $\beta$ and $\delta$ allows us to balance the influence of these components, yielding the structured objective

$$\mathcal{L}_{\theta,\phi}^{\text{FVAE}}(\mathbf{x}) = \alpha \mathop{\mathbb{E}}_{\mathbf{z}_1,\mathbf{z}_2} \left[\log p_\theta(\mathbf{x}|\mathbf{z}_1,\mathbf{z}_2)\right] - \beta D_{\text{KL}}\left(q_{\phi_1}(\mathbf{z}_1|\mathbf{x}) \,\|\, p_1(\mathbf{z}_1)\right) + \delta \underbrace{\mathop{\mathbb{E}}_{\mathbf{z}_2,\mathbf{z}_1} \log \frac{p_2(\mathbf{z}_2)}{p_1(\mathbf{z}_1)}}_{\Gamma}. \tag{4}$$

The second term correspond to the $D_{KL}$ in the $\beta$-VAE objective, ensuring that the main latent variable $\mathbf{z}_1$ captures the relevant information for reconstructing $\mathbf{x}$ while remaining close to its prior. The third term, $\Gamma$, acts as a repulsive regularizer, encouraging the second component $\mathbf{z}_2$ to decouple from $\mathbf{z}_1$. Note that, a priori, we could fix $\alpha = 1$ and use only $\beta$ and $\delta$ to weight the contributions of all the terms. However, we prefer to use all three, as it will make the interpretation of each contribution clearer later on.

## 3.3 Mechanism of the $\Gamma$ modulator

$\Gamma$ introduces an indirect interaction between the two encoders by modulating their alignment with their respective priors. Rather than enforcing separation through a direct divergence between posteriors, it shifts their latent support via prior-based regularization. To analyze the effect of $\Gamma$, we first rewrite it as the sum of a mismatch term and a discrepancy term, i.e.,

$$\Gamma = \underbrace{\mathbb{E}_{\mathbf{z}_2 \sim q_{\phi_2}}\left[\log p_2 - \log p_1\right]}_{\text{mismatch: } \Lambda} + \underbrace{\left[\mathbb{E}_{\mathbf{z}_2 \sim q_{\phi_2}} \log p_1 - \mathbb{E}_{\mathbf{z}_1 \sim q_{\phi_1}} \log p_1\right]}_{\text{discrepancy: } \Delta}, \tag{5}$$

where the mismatch term can be further equivalently expressed as a difference of KLs, i.e., $\Lambda = D_{KL}(q_{\phi_2}\|p_1) - D_{KL}(q_{\phi_2}\|p_2)$. This decomposition reveals a meaningful structure, as outlined in the following.

Maximizing the mismatch encourages $q_{\phi_2}$ to align with its prior $p_2$. This mirrors the behavior expected in a two-variable standard VAE, where each encoder is regularized toward its respective prior. As a result, we retain effective control over the behavior of $q_{\phi_2}$, providing a structural safeguard against degenerate or unconstrained posterior collapse. In contrast, it disincentivizes $q_{\phi_2}$ from aligning with the prior $p_1$. Consequently, $q_{\phi_2}$ is encouraged to preserve or discover features and structures that are distinct from, and not merely reflections of, the assumptions embedded within $p_1$. The mismatch term also highlights that the two priors $p_1$ and $p_2$ should be different, but still partially overlapping. If they are identical, some terms will simply cancel out, and if they are too far apart, the separation becomes trivial, resulting in no fruitful competition for occupying the latent space. Since the priors are usually Gaussian with variance 1, this competition is parameterized by $|\mu_1 - \mu_2|$; the effect of the mismatch is null when this parameter is null.

In addition, we can demonstrate (see Appendix B.1) that the discrepancy $\Delta$ is bounded by a term depending on the 2-Wasserstein distance, provided that the Hessian $\|\nabla^2 \log p_1\| \leq L$ is bounded. In practice, $p_1$ is typically a standard normal $\mathcal{N}(0, I)$, and $q_{\phi_1}$ is a diagonal Gaussian. Under these assumptions, the bound becomes:

$$\Delta \leq \frac{L}{2}\mathcal{W}_2^2(q_{\phi_1}, q_{\phi_2}) + \sqrt{\sum_j \mu_j^2 + \sigma_j^2} \cdot \mathcal{W}_2(q_{\phi_1}, q_{\phi_2}),$$

where $\mu_j$ and $\sigma_j^2$ are the parameters of $q_{\phi_1}$. Since $q_{\phi_1}$ is typically optimized to approximate $p_1$, the square-root term remains bounded in most settings. Both terms in the bound grow with $\mathcal{W}_2(q_{\phi_1}, q_{\phi_2})$, making $\Delta$ an effective surrogate for inducing Wasserstein repulsion. In particular, maximizing $\Delta$ increases $\mathcal{W}_2(q_{\phi_1}, q_{\phi_2})$, driving the two encoders apart in a geometrically meaningful way.

## 3.4  FVAE-LoRA

Building upon the FVAE framework, we leverage its ability to split the latent space to gain finer control over the representation, ultimately achieving better performance. To accomplish this, we proceed as illustrated in Figure 1.

For each targeted linear layer, we train an FVAE simultaneously with the downstream task, aiming to replace the $\mathbf{A}$ matrices used in classical LoRA (see the left side of the figure). During training, the input to the target layer is fed into the FVAE to compute a reconstruction loss based on that input. In parallel, the latent embedding $\mathbf{z}_1$ produced by the encoder $q_{\phi_1}$ is passed through a learned matrix $\mathbf{B}$ and added to the output of the frozen base weights $\mathbf{W}$. This yields the output $\mathbf{W}\mathbf{x} + \mathbf{B}\mathbf{z}_1$. At inference time, only $q_{\phi_1}$ is used to produce the output, either by sampling from it or by taking the mean of the distribution. Note that while we propose using FVAE with LoRA, the method is generic in the sense that it can be applied to give latent space control to any explicit LoRA method. In summary, the loss to be optimized in the proposed FVAE-LoRA approach is given by

$$\min_{\phi,\theta} \mathcal{L}_{\text{downstream-task}} - \boldsymbol{\lambda} \sum_{l \in \text{layer}} \mathcal{L}_{\theta,\phi}^{\text{FVAE}}(\mathbf{x}_l), \tag{6}$$

with $\boldsymbol{\lambda}$ being the hyper-parameter vector of weights assigned to the FVAE loss in each layer.

**In practice:** Both $q_{\phi_1}$ and $q_{\phi_2}$ are parameterized as diagonal Gaussian distributions, with their means and variances learned by neural networks. The reconstruction term $p_\theta(\mathbf{x}|\mathbf{z}_1, \mathbf{z}_2)$ is also parameterized by a neural network. The prior $p_1 = \mathcal{N}(\mathbf{0}, \mathbf{I})$ is a standard normal distribution, while $p_2$ is empirically chosen to be centered at $\mathbf{1.5}$. The intuition is to give the two priors distinct non-overlapping "location" in the latent space to initialize and encourage separation. By setting $\mu_1$ at $0$ and $\mu_2$ at $1.5$, we provide a clear signal for the repulsive regularizer to push the posteriors apart. See additional insights in Appendices E and F.

# 4  Experimental Results

**Motivation.**  The objective of the experimental evaluation is two fold. First, we aim to comprehensively evaluate FVAE-LoRA by comparing its performance against standard LoRA and its relevant variants across diverse image, text, and audio tasks. The specific selection of relevant variants for each domain is detailed within the respective modality subsections, guided by the aim to provide the most insightful and relevant benchmarks for each specific context. Second, we seek to empirically validate that FVAE-LoRA learns more robust representations by preferentially encoding task-salient information in $\mathbf{z}_1$.

**Overall Setup.**  To ensure fair comparisons of parameter efficiency for the core adaptation mechanism, the LoRA rank $r$ is set to $16$ for all LoRA-based methods throughout our experiments. This rank also corresponds to the dimensionality of the task-salient latent space $\mathbf{z}_1$ in FVAE-LoRA. All LoRA-based baselines as well as FVAE-LoRA are applied to the query and key matrices within the transformer models. Detailed hyperparameter settings for FVAE-LoRA, including the balancing coefficients $\alpha$, $\beta$ and $\delta$, learning rates, and specific VAE architectural choices for each task, are provided in Appendix D. We also provide a practical guide for selecting the key factorization hyperparameters, $\beta$ and $\delta$, in Appendix G.

## 4.1  Efficacy of FVAE-LoRA for Various Downstream Tasks

### 4.1.1  Image Tasks

**Datasets.**  We evaluate FVAE-LoRA on six diverse image classification datasets: DTD [32], EuroSAT [33], GTSRB [34], RESISC45 [35], SUN397 [36], and SVHN [37]. These datasets span various image types, domains, and complexities.

**Implementation Details.** The pre-trained Vision Transformer (ViT-B/16) [3] serves as the back-bone model for all image classification tasks. We compare FVAE-LoRA against full fine-tuning (Full FT) and several LoRA variants, i.e., standard LoRA [12], PiSSA [20], rsLoRA [21], DoRA [15], and OLoRA [38]. This broad selection of LoRA variants represents established PEFT methods for image classification. The evaluation metric is top-1 accuracy. Detailed hyperparameters can be found in D.1.

Table 1: Fine-tuning results of ViT-B/16 on image classification tasks. We fine-tune ViT-B/16 using full fine-tuning and LoRA variants across DTD, EuroSAT, GTSRB, RESISC45, SUN397, and SVHN datasets. **Bold** indicates the highest performance, while underline represents the second-highest performance.

| Method | Params (%) | DTD | EuroSAT | GTSRB | RESISC45 | SUN397 | SVHN | Average |
|---|---|---|---|---|---|---|---|---|
| Full FT | - | 78.12±0.59 | **98.30±0.47** | **98.85±0.14** | **94.35±0.54** | 69.34±0.59 | **97.34±0.03** | 89.38 |
| LoRA | 0.7240 | 74.65±1.08 | 97.28±0.36 | 96.95±0.56 | 90.11±0.53 | 71.11±0.07 | 94.22±0.14 | 87.39 |
| PiSSA | 0.7240 | 74.22±1.69 | 97.33±0.31 | 96.95±1.28 | 89.82±0.37 | 69.09±0.19 | 94.83±0.73 | 87.04 |
| rsLoRA | 0.7240 | 72.23±1.00 | 97.48±0.21 | 96.63±0.83 | 88.04±0.30 | 67.69±0.32 | 93.81±0.77 | 85.98 |
| DoRA | 0.7451 | 75.74±1.91 | 97.28±0.80 | 97.27±0.44 | 91.72±1.17 | 71.53±0.25 | 96.41±0.72 | 88.32 |
| OLoRA | 0.7240 | 72.23±0.40 | 96.62±1.06 | 97.08±0.46 | 88.94±0.45 | 69.64±0.43 | 94.86±0.30 | 86.63 |
| FVAE-LoRA | 0.7311 | **78.19±0.68** | 97.78±0.15 | 97.98±0.56 | 93.57±0.22 | **73.14±0.21** | 96.55±0.05 | **89.53** |

**Results.** The effectiveness of FVAE-LoRA for image classification is shown in Table 1. FVAE-LoRA achieves an average accuracy of 89.53% across six diverse datasets, outperforming LoRA and surpassing variants such as DoRA, all within a comparable inference-time parameter budget.

Notably, FVAE-LoRA's average performance slightly surpasses that of full fine-tuning (89.38%). This result suggests that the structured latent factorization inherent to FVAE-LoRA can guide the model towards learning highly effective adaptations. By explicitly encouraging the disentanglement of task-salient information within $z_1$, FVAE-LoRA might be more adept at focusing the ViT backbone on critical visual features for the downstream task, potentially mitigating the risk of overfitting to spurious correlations or less generalizable patterns that can sometimes affect full fine-tuning on these datasets. On challenging datasets such as DTD (characterized by fine-grained textures) and SUN397 (complex scenes), FVAE-LoRA particularly excels, achieving the highest scores and outperforming full fine-tuning. For instance, on SUN397, FVAE-LoRA demonstrates a clear advantage, indicative of its capacity to distill critical visual cues for complex recognition tasks. While full fine-tuning outperforms all LoRA variants on datasets like EuroSAT and GTSRB, FVAE-LoRA consistently stands as the leading or a highly competitive PEFT method, often closing the gap significantly (e.g., achieving 97.78% on EuroSAT, closely trailing Full FT's 98.30%).

The presented results show that FVAE-LoRA is able to learn highly effective low-rank updates through a principled approach to information selection.

### 4.1.2 Text Tasks

**Datasets.** For natural language tasks, we use two benchmark categories:

1. **Commonsense Reasoning**: Training is done on a predefined corpus [39][1] of query-answer pairs, and the evaluation set includes seven sub-tasks: PIQA [40] (physical commonsense), SIQA [41] (social interaction understanding), ARC-c and ARC-e [42] (science question answering), OBQA [43] (multi-hop reasoning over facts), HellaSwag [44] (commonsense natural language inference)), and WinoGrande [45] (fill-in-the-blank).

2. **GLUE Benchmark**: A subset of the GLUE [46] is used, comprising SST2 (sentiment analysis), CoLA (linguistic acceptability), QNLI (question-answering NLI), MRPC (para-phrase detection), RTE (textual entailment), STSB (semantic textual similarity), and WNLI (coreference resolution).

**Implementation Details.** We employ Llama-3-8B [8] for the commonsense reasoning tasks and roberta-base [47] for the GLUE benchmark tasks. For commonsense reasoning tasks, we compare against Prompt Tuning [11], P-Tuning [10, 48], standard LoRA, and HiRA [13]. For completeness,

---

[1] https://github.com/AGI-Edgerunners/LLM-Adapters/tree/main/dataset

we also present the performance of ChatGPT taken from [15]. Considering the computational cost of LLM fine-tuning, our LoRA-based comparisons focus on standard LoRA and HiRA, as HiRA has recently demonstrated strong performance, offering a relevant and challenging benchmark in this setting. For roberta-base on GLUE, comparisons are made against Full FT and standard LoRA. This allows for a direct assessment of FVAE-LoRA's parameter efficiency relative to the crucial full fine-tuning upper bound and the widely adopted LoRA baseline. Evaluation uses accuracy for commonsense tasks following [13] and standard GLUE metrics (Matthews Correlation for CoLA, Pearson Correlation for STSB, Accuracy for the rest). Detailed hyperparameters can be found in D.2.

Table 2: Accuracy comparison among various PEFT methods on commonsense reasoning datasets for Llama-3-8B. **Bold** indicates the best performance, while underline represents the second-best performance. ChatGPT performance values are taken from [15], whereas Prompt Tuning and P-Tuning from [13].

| Model | Method | Params (%) | PIQA | SIQA | ARC-c | ARC-e | OBQA | HellaSwag | WinoGrande | Average |
|---|---|---|---|---|---|---|---|---|---|---|
| ChatGPT | - | - | 85.40 | 68.50 | 79.90 | 89.80 | 74.80 | 78.50 | 66.10 | 77.57 |
| | Prompt Tuning | 0.0010 | 45.05 | 36.13 | 31.57 | 32.74 | 29.20 | 14.01 | 50.12 | 34.12 |
| | P-Tuning | 0.6240 | 11.64 | 8.19 | 7.42 | 8.63 | 9.60 | 1.77 | 37.65 | 12.13 |
| Llama-3-8B | LoRA | 0.0848 | 80.74 | 75.59 | 67.58 | 82.11 | 75.20 | 85.73 | 77.82 | 77.82 |
| | HiRA | 0.0848 | 88.63 | 80.40 | **81.66** | **93.56** | **87.20** | 94.48 | 85.87 | 87.40 |
| | FVAE-LoRA | 0.0850 | **88.96** | **81.58** | 81.06 | 92.72 | 86.20 | **95.30** | **88.95** | **87.82** |

**Results on Commonsense Reasoning using Llama-3-8B model.** Table 2 reports the performance of FVAE-LoRA and baselines across seven commonsense reasoning benchmarks using the LLaMA-3-8B model. Our approach achieves the highest average accuracy of 87.82%, outperforming both the strong HiRA baseline (87.40%) and LoRA (77.82%) under comparable inference-time parameter budgets. These results indicate that FVAE-LoRA's strategy of factorizing latent information is particularly beneficial for complex reasoning tasks in LLMs. By explicitly guiding the $z_1$ latent space to capture task-salient semantic and contextual cues necessary for reasoning, FVAE-LoRA enables Llama-3-8B to make more accurate inferences.

Notably, FVAE-LoRA demonstrates strong individual performances on tasks like HellaSwag (95.30%) and WinoGrande (88.95%), which require nuanced understanding of everyday situations and disambiguation. This suggests that the information isolated in $z_1$ is indeed critical for these types of reasoning, allowing the LLM to leverage its capabilities more effectively than with less structured adaptation techniques. The ability to improve upon already powerful models like Llama-3-8B with such parameter efficiency highlights the potential of FVAE-LoRA for targeted capability enhancement in large language models.

Table 3: Results of fine-tuning roberta-base using full fine-tuning and LoRA on a subset of the GLUE datasets. **Bold** indicates the best results, while underline represents the second-best results.

| Method | Params (%) | SST2 | CoLA | QNLI | MRPC | RTE | STSB | WNLI | Average |
|---|---|---|---|---|---|---|---|---|---|
| Full FT | - | **94.77±0.25** | **62.43±1.16** | **91.97±0.05** | 89.40±0.70 | 79.53±1.31 | **90.30±0.29** | 56.30±0.00 | 80.67 |
| LoRA | 0.4710 | 93.97±0.47 | 59.60±1.02 | 91.87±0.19 | 88.73±0.61 | 77.87±0.74 | 88.90±0.45 | 57.73±2.03 | 79.81 |
| FVAE-LoRA | 0.4759 | 94.10±0.43 | 60.37±0.49 | 91.63±0.34 | **89.53±0.97** | **79.90±0.85** | 88.60±0.16 | **64.33±2.38** | **81.21** |

**Results on GLUE benchmark.** Table 3 presents the performance of FVAE-LoRA when adapting the roberta-base model on a subset of the GLUE benchmark. Our method achieves the highest average score (81.21), outperforming both full fine-tuning (80.67) and standard LoRA (79.81). Notably, FVAE-LoRA shows particular strength on tasks like MRPC and WNLI. This strong performance on roberta-base demonstrates that the benefits of FVAE-LoRA's explicit latent factorization are not confined to large-scale models like Llama-3-8B (as seen in commonsense reasoning tasks), but also translate effectively to smaller, yet widely utilized encoder models. The ability to enhance these more moderately-sized architectures suggests that FVAE-LoRA's principled approach to focusing adaptations via $z_1$ on task-critical linguistic features is robust across different model scales.

### 4.1.3  Audio Tasks

**Datasets.** We conduct automatic speech recognition (ASR) on the TIMIT acoustic-phonetic corpus [49] for phoneme recognition.

**Implementation Details.** The pre-trained Wav2Vec2-Large model [5] serves as the backbone. Fine-tuning utilizes the Connectionist Temporal Classification loss [50]. We compare against Full FT and standard LoRA. Performance is measured by Phoneme Error Rate (PER). Detailed hyperparameters are provided in Appendix D.3.

**Results.** As shown in Table 4, FVAE-LoRA achieves a PER of 8.09 on TIMIT, outperforming standard LoRA and approaching the performance of full fine-tuning (7.48), demonstrating its effectiveness for ASR.

Table 4: Fine-tuning results of Wav2Vec2-Large on the TIMIT speech recognition task using CTC loss. **Bold** indicates the best PER ($\downarrow$), underline the second-best.

| Method | Params (%) | TIMIT PER ($\downarrow$) |
|---|---|---|
| Full FT | - | **7.48** |
| LoRA | 0.4961 | 9.38 |
| FVAE-LoRA | 0.4999 | 8.09 |

## 4.2 Probing Latent Factorization via Spurious Correlation

To empirically validate our hypothesis that FVAE-LoRA learns more robust representations by preferentially encoding task-salient information in $z_1$, we conduct experiments using datasets with controlled spurious correlations. Spurious correlations occur when input features are statistically associated with target labels without a true causal link [51, 52, 53], potentially misleading models and hindering generalization, especially on out-of-distribution or minority-group data. Our aim is to assess whether FVAE-LoRA's disentanglement mechanism renders it more robust to such misleading cues compared to standard LoRA.

**Experimental Design.** We leverage datasets where spurious attributes (e.g., background scene) are intentionally correlated with the true class labels (e.g., object category) during training. For example, a "landbird" might predominantly appear against a "land" background, and a "waterbird" against "water". Effective factorization should enable the model to learn the true object category via $z_1$, irrespective of the potentially misleading background. Figure 2 illustrates this concept, distinguishing between an input image ($x$), its core features ($x_{core}$), and its spurious features ($x_{spurious}$).

**Datasets.** Following prior works [54, 51, 52, 53, 55, 56], we consider three standard benchmarks to introduce spurious correlations: *Waterbirds* [56], where bird type (landbird vs. waterbird) is correlated with background (land vs. water); *CelebA* [56], where a target attribute (e.g., blonde hair) might be correlated with another attribute (e.g., being female); and *Animals* [57], a larger-scale dataset derived from ImageNet [58] with four animal classes spuriously correlated with background types (e.g., waterbirds with water, small dogs with indoor scenes). These datasets are structured into groups based on combinations of true labels and spurious attributes, with varying majority-to-minority group ratios between training and test splits (details in Appendix C.1 and Table 7).

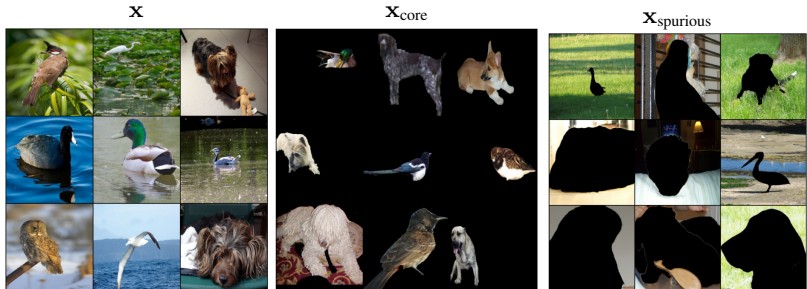

Figure 2: Random samples drawn from the train split of the Animals dataset, illustrating an original image ($x$), its core object features ($x_{core}$), and its spurious background features ($x_{spurious}$).

**Implementation Details and Evaluation Metrics.** We adapt the ViT-B/16 [3] backbone using LoRA and our proposed FVAE-LoRA. Following common practice in literature [59, 60, 61], performance is evaluated using three key metrics:

- **Worst-Group Accuracy (WG):** Accuracy on the test subgroup where the model performs poorest, indicating robustness to spurious correlations and performance on minority groups.

- **Average Accuracy (AVG):** Standard overall accuracy on the test set.

- **Accuracy Disparity:** The absolute difference |WG − AVG|, quantifying the performance variation across groups. A smaller disparity suggests more uniform and equitable performance.

Table 5: Fine-tuning results of ViT-B/16 on spurious correlation benchmarks. We compare LoRA with FVAE-LoRA on ANIMALS (8 groups, 4 classes), WATERBIRDS (4 groups, 2 classes), and CELEBA (4 groups, 2 classes) datasets.

| Method | Params (%) | ANIMALS | | WATERBIRDS | | CELEBA | | Disparity |
|---|---|---|---|---|---|---|---|---|
| | | WG | AVG | WG | AVG | WG | AVG | \| WG - AVG \| |
| LoRA | 0.7240 | 54.79±8.08 | 88.20±1.17 | 75.49±0.9 | 90.39±0.78 | 40.00±3.54 | **96.09±0.02** | 34.8 |
| FVAE-LoRA | 0.7311 | **62.0±4.83** | **89.55±0.96** | **75.85±3.72** | **90.99±0.51** | **43.33±6.68** | 95.77±0.18 | 31.71 |

**Results.** Table 5 summarizes the performance of standard LoRA, and FVAE-LoRA on the spurious correlation benchmarks. Across all datasets, FVAE-LoRA consistently achieves higher WG and lower Accuracy Disparity compared to LoRA, while maintaining competitive AVG. These findings strongly suggest that FVAE-LoRA is less susceptible to being misled by spurious features present in the training data. We attribute this enhanced robustness to the explicit factorization encouraged by our novel ELBO. By compelling $z_1$ to capture genuinely task-relevant, causal features and relegating other variations, FVAE-LoRA learns a more robust adaptation. This leads to improved generalization, particularly on minority groups where spurious cues are often unreliable or reversed, thereby validating the intended robust learning mechanism of our proposed method.

Table 6: Ablation study comparing our FVAE-LoRA when fine-tuning ViT-B/16 to a two-latent-variable VAE (VAE2LAT, as defined in Eq. (2)), and $\beta$-VAE2LAT, the $\beta$-VAE version of VAELAT (where all the DKL terms are multiplied by 10). Results are presented on DTD, EuroSAT, GTSRB, RESISC45, SUN397, and SVHN. **Bold** indicates the highest results, while underlined indicates the second-highest.

| Method | DTD | EuroSAT | GTSRB | RESISC45 | SUN397 | SVHN | Average |
|---|---|---|---|---|---|---|---|
| FVAE-LoRA (Proposed) | **78.19±0.68** | **97.78±0.15** | **97.98±0.56** | **93.57±0.22** | **73.14±0.21** | **96.55±0.05** | **89.53** |
| $\beta$-VAE2LAT (8) (where $\beta = 10$) | 77.16±0.43 | 96.86±0.15 | 95.75±0.46 | 89.46±0.13 | 72.91±0.32 | 91.58±0.69 | 87.29 |
| VAE2LAT (3) | 75.96±0.80 | 96.64±0.59 | 94.38±0.94 | 88.42±0.32 | 71.68±0.54 | 91.50±0.58 | 86.43 |

## 4.3 Ablation Studies

To demonstrate the relevance of introducing the regularization term in Equation ((3)), we replicate our image results using the two-variable VAE model (2) and its equivalent for $\beta$-VAE with two latent variables (where the two KL divergences have been multiplied by 10; see Equation (A.3)). The results can be seen in Table 6. The baseline model performs the worst across all datasets. The $\beta$-VAE with two latent variables shows some improvement, but it is still outperformed by our proposed method.

## 5 Conclusions

We introduced Factorized Variational Autoencoder LoRA (FVAE-LoRA), a novel PEFT method designed to explicitly disentangle task-salient information within the LoRA framework. By employing a VAE with two latent spaces, $z_1$ (task-salient) and $z_2$ (residual), and a specialized ELBO, FVAE-LoRA ensures that the adaptive updates are primarily driven by task-critical features learned in $z_1$. Our comprehensive evaluations on diverse text, audio, and image benchmarks demonstrated that FVAE-LoRA consistently surpasses standard LoRA in performance. Crucially, experiments on datasets with spurious correlations empirically confirmed that FVAE-LoRA's factorization leads to more robust representations, as evidenced by improved worst-group accuracy. FVAE-LoRA highlights the potential of latent space factorization for enhancing parameter-efficient fine-tuning.

# 6 Acknowledgments

Shashi Kumar was partially supported by the EU Horizon 2020 project ELOQUENCE (grant number 101070558). Yacouba Kaloga was partially supported by Swiss National Science Foundation project no CRSII5_202228 on Characterisation of motor speech disorders and processes.

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

# A  Variational Auto-Encoder

## A.1  VAE Objective Derivation

We derive the Evidence Lower Bound (ELBO) by starting from the marginal log-likelihood:

$$
\begin{aligned}
\log p_\theta(\mathbf{x}) &= \mathbb{E}_{\mathbf{z}\sim q_\phi(\mathbf{z}|\mathbf{x})}\left[\log p_\theta(\mathbf{x})\right]\\
&= \mathbb{E}_{\mathbf{z}\sim q_\phi(\mathbf{z}|\mathbf{x})}\left[\log\left(\frac{p_\theta(\mathbf{x}|\mathbf{z})p(\mathbf{z})}{p_\theta(\mathbf{z}|\mathbf{x})}\right)\right]\\
&= \mathbb{E}_{\mathbf{z}\sim q_\phi(\mathbf{z}|\mathbf{x})}\left[\log p_\theta(\mathbf{x}|\mathbf{z}) + \log\frac{p(\mathbf{z})}{q_\phi(\mathbf{z}|\mathbf{x})} + \log\frac{q_\phi(\mathbf{z}|\mathbf{x})}{p_\theta(\mathbf{z}|\mathbf{x})}\right]\\
&= \mathbb{E}_{\mathbf{z}\sim q_\phi(\mathbf{z}|\mathbf{x})}\left[\log p_\theta(\mathbf{x}|\mathbf{z})\right] - D_{\mathrm{KL}}\left(q_\phi(\mathbf{z}|\mathbf{x})\,\|\,p(\mathbf{z})\right) + D_{\mathrm{KL}}\left(q_\phi(\mathbf{z}|\mathbf{x})\,\|\,p_\theta(\mathbf{z}|\mathbf{x})\right).
\end{aligned}
$$

The last term is always non-negative, which justifies interpreting the remaining two terms as a lower bound, i.e.,

$$
\log p_\theta(\mathbf{x}) \geq \mathcal{L}^{\mathrm{VAE}}_{\theta,\phi}(\mathbf{x}),
$$

with

$$
\mathcal{L}^{\mathrm{VAE}}_{\theta,\phi}(\mathbf{x}) = \mathbb{E}_{\mathbf{z}\sim q_\phi(\mathbf{z}|\mathbf{x})}\left[\log p_\theta(\mathbf{x}|\mathbf{z})\right] - D_{\mathrm{KL}}\left(q_\phi(\mathbf{z}|\mathbf{x})\,\|\,p(\mathbf{z})\right).
$$

## A.2  VAE2LAT: VAE with 2 Latent Variables Objective Derivation

We simply start from the ELBO previously derived with two variables, i.e.,

$$
\mathcal{L}^{\mathrm{VAE2LAT}}_{\theta,\phi}(\mathbf{x}) = \mathbb{E}_{\mathbf{z}\sim q_\phi(\mathbf{z}_1,\mathbf{z}_2|\mathbf{x})}\left[\log p_\theta(\mathbf{x}|\mathbf{z}_1,\mathbf{z}_2)\right] - D_{\mathrm{KL}}\left(q_\phi(\mathbf{z}_1,\mathbf{z}_2|\mathbf{x})\,\|\,p(\mathbf{z}_1,\mathbf{z}_2)\right).
$$

Applying the independence assumption, we obtain

$$
\mathcal{L}^{\mathrm{VAE2LAT}}_{\theta,\phi}(\mathbf{x}) = \mathbb{E}_{\substack{\mathbf{z}_1\sim q_{\phi_1}(\mathbf{z}_1|\mathbf{x})\\\mathbf{z}_2\sim q_{\phi_2}(\mathbf{z}_2|\mathbf{x})}}\left[\log p_\theta(\mathbf{x}|\mathbf{z}_1,\mathbf{z}_2)\right] - D_{\mathrm{KL}}\left(q_{\phi_1}(\mathbf{z}_1|\mathbf{x})\,\|\,p_1(\mathbf{z}_1)\right) - D_{\mathrm{KL}}\left(q_{\phi_2}(\mathbf{z}_2|\mathbf{x})\,\|\,p_2(\mathbf{z}_2)\right).
$$

$$(7)$$

## A.3  $\beta$-VAE2LAT: A $\beta$-VAE with 2 latents variables

The loss of $\beta$-VAE2LAT, i.e., a straightforward extension of $\beta$-VAE to two latent variables is given by studies 4.3.

$$
\mathcal{L}^{\boldsymbol{\beta}-\mathrm{VAE2LAT}}_{\theta,\phi}(\mathbf{x}) = \mathbb{E}_{\substack{\mathbf{z}_1\sim q_{\phi_1}(\mathbf{z}_1|\mathbf{x})\\\mathbf{z}_2\sim q_{\phi_2}(\mathbf{z}_2|\mathbf{x})}}\left[\log p_\theta(\mathbf{x}|\mathbf{z}_1,\mathbf{z}_2)\right] - \beta D_{\mathrm{KL}}\left(q_{\phi_1}(\mathbf{z}_1|\mathbf{x})\,\|\,p_1(\mathbf{z}_1)\right) - \beta D_{\mathrm{KL}}\left(q_{\phi_2}(\mathbf{z}_2|\mathbf{x})\,\|\,p_2(\mathbf{z}_2)\right).
$$

$$(8)$$

This formulation is used in the ablation studies in Section 4.3.

# B  FVAE

## B.1  Bounding the Discrepancy Term $\Delta$ via the 2-Wasserstein Distance

We bound the discrepancy

$$
\Delta = \mathbb{E}_{\mathbf{z}\sim q_{\phi_2}}[\log p_1(\mathbf{z})] - \mathbb{E}_{\mathbf{z}\sim q_{\phi_1}}[\log p_1(\mathbf{z})],
$$

assuming only that the log-prior $f(z) = \log p_1(z)$ is $C^2$ with a globally bounded Hessian:

$$
\|\nabla^2 f(z)\|_{\mathrm{op}} \leq L \quad \forall z \in \mathbb{R}^d.
$$

$$(\mathrm{H})$$

**Step 1 – Second-order Taylor control.** For any two points $z_1, z_2$, Taylor's formula with integral remainder gives

$$f(z_2) - f(z_1) = \langle \nabla f(z_1), z_2 - z_1 \rangle + (z_2 - z_1)^\top \left( \int_0^1 (1-s) \nabla^2 f(z_1 + s(z_2 - z_1)) ds \right)(z_2 - z_1).$$

Bounding the remainder using assumption (H) gives the point-wise inequality

$$\left| f(z_2) - f(z_1) - \langle \nabla f(z_1), z_2 - z_1 \rangle \right| \leq \frac{L}{2} \|z_2 - z_1\|^2. \tag{A}$$

**Step 2 – Integrate over a coupling.** Let $\gamma \in \Pi(q_{\phi_1}, q_{\phi_2})$ be *any* coupling of the two distributions, and write $(\mathbf{z}_1, \mathbf{z}_2) \sim \gamma$, $d = \mathbf{z}_2 - \mathbf{z}_1$. Taking expectations in (A), applying the triangle inequality, and then Cauchy–Schwarz to the linear term,

$$|\Delta| \leq \frac{L}{2} \mathbb{E}_\gamma \|d\|^2 + \underbrace{\left| \mathbb{E}_\gamma \langle \nabla f(\mathbf{z}_1), d \rangle \right|}_{\text{``linear term expectation''}} \leq \frac{L}{2} \mathbb{E}_\gamma \|d\|^2 + \sqrt{\mathbb{E}_{q_{\phi_1}} \|\nabla f\|^2} \sqrt{\mathbb{E}_\gamma \|d\|^2}.$$

Now minimise the rightmost expression over $\gamma$. Since the function $g(x) = \frac{L}{2} x^2 + Cx$ (for $C = \sqrt{\mathbb{E}_{q_{\phi_1}} \|\nabla f\|^2} \geq 0$) is non-decreasing for $x = \sqrt{\mathbb{E}_\gamma \|d\|^2} \geq 0$, the infimum is attained when $\mathbb{E}_\gamma \|d\|^2$ is minimized. The infimum of $\mathbb{E}_\gamma \|d\|^2$ is $\mathcal{W}_2^2(q_{\phi_1}, q_{\phi_2})$. Hence :

$$|\Delta| \leq \frac{L}{2} \mathcal{W}_2^2(q_{\phi_1}, q_{\phi_2}) + \sqrt{\mathbb{E}_{q_{\phi_1}} \|\nabla \log p_1(\mathbf{z})\|^2} \, \mathcal{W}_2(q_{\phi_1}, q_{\phi_2}).$$

**Step 3 – Specialization to Gaussian case.** Assume $p_1 = \mathcal{N}(0, I)$ and $q_{\phi_1} = \mathcal{N}(\boldsymbol{\mu}, \text{diag}(\boldsymbol{\sigma}^2))$. Then the gradient becomes $\nabla \log p_1(\mathbf{z}) = -\mathbf{z}$, and the expectation simplifies as:

$$\mathbb{E}_{q_{\phi_1}} \|\nabla \log p_1(\mathbf{z})\|^2 = \mathbb{E}_{q_{\phi_1}} \left[ \|\mathbf{z}\|^2 \right] = \sum_j \mu_j^2 + \sigma_j^2.$$

Therefore, the bound becomes:

$$|\Delta| \leq \frac{1}{2} \mathcal{W}_2^2(q_{\phi_1}, q_{\phi_2}) + \sqrt{\sum_j \mu_j^2 + \sigma_j^2} \cdot \mathcal{W}_2(q_{\phi_1}, q_{\phi_2}).$$

Since $q_{\phi_1}$ is trained to approximate $p_1$, the square-root term is typically bounded in practice. Hence, both terms contribute to increasing $\mathcal{W}_2(q_{\phi_1}, q_{\phi_2})$, and the discrepancy $\Delta$ serves as an effective Wasserstein repulsion.

# C   Dataset Details

## C.1   Spurious Correlation Experiments

Table 7: Statistics of the datasets used in the spurious experiment.

| Dataset | SpuCoAnimals | Waterbirds | CelebA |
|---|---|---|---|
| # Classes | 4 | 2 | 2 |
| # Groups | 8 | 4 | 4 |
| Train | 42000 | 4795 | 162770 |
| Validation | 2100 | 1199 | 19867 |
| Test | 4000 | 5794 | 19962 |
| Class Ratio | 25:25:25:25 | 76.8:23.2 | 85:15 |

# D   Hyperparameters

This section details the hyperparameters used for the experiments presented in the main paper. For all LoRA-based methods, including FVAE-LoRA, the LoRA rank ($r$) was set to 16, and LoRA was applied to the query and key matrices of the attention layers. The latent dimension of $\mathbf{z}_1$ in FVAE-LoRA corresponds to this LoRA rank.

### D.1 Image Experiments

The following hyperparameters were used for fine-tuning ViT-B/16 on DTD, EuroSAT, GTSRB, RESISC45, SUN397, and SVHN datasets.

Table 8: Hyperparameters for Image Classification tasks using ViT-B/16.

| Parameter | Value / Setting |
|---|---|
| *General Training Parameters* | |
| Optimizer | AdamW |
| Learning Rate | $5 \times 10^{-3}$ |
| LR Scheduler | Linear |
| Warmup Ratio | 0.1 |
| Batch Size | 32 |
| Number of Epochs | 30 |
| Weight Decay | 0.01 |
| Seeds | 1, 2, 42 |
| *LoRA Parameters* | |
| LoRA Rank ($r$) | 16 |
| LoRA Dropout | 0.1 |
| *FVAE-LoRA Specific Parameters* | |
| Latent Dim. $\mathbf{z}_1$ | 16 (same as LoRA rank) |
| Latent Dim. $\mathbf{z}_2$ | 16 |
| FVAE $q_{\phi_i}(\mathbf{z}_i|\mathbf{x})$ Enc. Arch. | $\mathbf{x} \xrightarrow{\text{Linear}} \dim(\mathbf{z}_i) \xrightarrow{\text{ReLU}} \text{HiddenState}_{\mathbf{z}_i} \xrightarrow{\text{Linear}} (\boldsymbol{\mu}_{\mathbf{z}_i}, \log \boldsymbol{\sigma}^2_{\mathbf{z}_i})$ |
| FVAE $p_\theta(\mathbf{x}|\mathbf{z}_1, \mathbf{z}_2)$ Dec. Arch. | $\text{Concat}(\mathbf{z}_1, \mathbf{z}_2) \xrightarrow{\text{Linear}} H_D = 128 \xrightarrow{\text{ReLU}} \text{Linear} \rightarrow \hat{\mathbf{x}} \text{ (Input Dim)}$ |
| Prior $p_1(\mathbf{z}_1)$ | $\mathcal{N}(0, I)$ |
| Prior $p_2(\mathbf{z}_2)$ | $\mathcal{N}(1.5, I)$ |
| $\boldsymbol{\lambda}$ (Eq. 6) | $1 \times 10^{-3}$ |
| ELBO Coeff. $\alpha$ (Reconstr.) | 1 |
| ELBO Coeff. $\beta$ (KL $q_1 \| p_1$) | 1 or 10 |
| ELBO Coeff. $\delta$ | 1 |

## D.2 Text Experiments

Table 9: Hyperparameters for Commonsense Reasoning using Llama-3-8B.

| Parameter | Value / Setting |
|---|---|
| *General Training Parameters* | |
| Optimizer | AdamW |
| Learning Rate | $1 \times 10^{-3}$ (LoRA, HiRA), $3 \times 10^{-4}$ (FVAE-LoRA) |
| LR Scheduler | Linear |
| Warmup Steps | 100 |
| Batch Size | 8 |
| Gradient Accumulation Steps | 4 |
| Number of Epochs | 3 |
| Weight Decay | 0.0 |
| Seed | 42 |
| *LoRA Parameters* | |
| LoRA Rank ($r$) | 16 |
| LoRA Dropout | 0.1 |
| Target Modules | q_proj, k_proj |
| *FVAE-LoRA Specific Parameters* | |
| Latent Dim. $\mathbf{z}_1$ | 16 |
| Latent Dim. $\mathbf{z}_2$ | 16 |
| FVAE $q_{\phi_i}(\mathbf{z}_i|\mathbf{x})$ Enc. Arch. | $\mathbf{x} \xrightarrow{\text{Linear}} \dim(\mathbf{z}_i) \xrightarrow{\text{ReLU}} \text{HiddenState}_{\mathbf{z}_i} \xrightarrow{\text{Linear}} (\boldsymbol{\mu}_{\mathbf{z}_i}, \log \boldsymbol{\sigma}^2_{\mathbf{z}_i})$ |
| FVAE $p_\theta(\mathbf{x}|\mathbf{z}_1, \mathbf{z}_2)$ Dec. Arch. | $\text{Concat}(\mathbf{z}_1, \mathbf{z}_2) \xrightarrow{\text{Linear}} H_D = 128 \xrightarrow{\text{ReLU}} \text{Linear} \to \hat{\mathbf{x}} \text{ (Input Dim)}$ |
| Prior $p_1(\mathbf{z}_1)$ | $\mathcal{N}(0, I)$ |
| Prior $p_2(\mathbf{z}_2)$ | $\mathcal{N}(1.5, I)$ |
| $\boldsymbol{\lambda}$ (Eq. 6) | $1 \times 10^{-4}$ |
| ELBO Coeff. $\alpha$ (Reconstr.) | 1 |
| ELBO Coeff. $\beta$ (KL $q_1||p_1$) | 1 or 10 |
| ELBO Coeff. $\delta$ | 1 |

Table 10: Hyperparameters for GLUE Benchmark tasks using RoBERTa-base.

| Parameter | Value / Setting |
|---|---|
| *General Training Parameters* | |
| Optimizer | AdamW |
| Learning Rate | $3 \times 10^{-4}$ |
| LR Scheduler | Linear |
| Warmup Ratio | 0.06 |
| Batch Size | 32 |
| Number of Epochs | 30 |
| Seed | 1, 2, 42 |
| *LoRA Parameters* | |
| LoRA Rank ($r$) | 16 |
| LoRA Dropout | 0.1 |
| *FVAE-LoRA Specific Parameters* | |
| Latent Dim. $\mathbf{z}_1$ | 16 |
| Latent Dim. $\mathbf{z}_2$ | 16 |
| FVAE $q_{\phi_i}(\mathbf{z}_i|\mathbf{x})$ Enc. Arch. | $\mathbf{x} \xrightarrow{\text{Linear}} \dim(\mathbf{z}_i) \xrightarrow{\text{ReLU}} \text{HiddenState}_{\mathbf{z}_i} \xrightarrow{\text{Linear}} (\boldsymbol{\mu}_{\mathbf{z}_i}, \log \boldsymbol{\sigma}^2_{\mathbf{z}_i})$ |
| FVAE $p_\theta(\mathbf{x}|\mathbf{z}_1, \mathbf{z}_2)$ Dec. Arch. | $\text{Concat}(\mathbf{z}_1, \mathbf{z}_2) \xrightarrow{\text{Linear}} H_D = 128 \xrightarrow{\text{ReLU}} \text{Linear} \to \hat{\mathbf{x}} \text{ (Input Dim)}$ |
| Prior $p_1(\mathbf{z}_1)$ | $\mathcal{N}(0, I)$ |
| Prior $p_2(\mathbf{z}_2)$ | $\mathcal{N}(1.5, I)$ |
| $\boldsymbol{\lambda}$ (Eq. 6) | $1 \times 10^{-3}$ or $1 \times 10^{-4}$ |
| ELBO Coeff. $\alpha$ (Reconstr.) | 0.1 or 1 |
| ELBO Coeff. $\beta$ (KL $q_1||p_1$) | 1 or 10 |
| ELBO Coeff. $\delta$ | 1 |

## D.3 Audio Experiments

The following hyperparameters were used for fine-tuning Wav2Vec2-Large on the TIMIT dataset.

Table 11: Hyperparameters for ASR on TIMIT using Wav2Vec2-Large.

| Parameter | Value / Setting |
|---|---|
| *General Training Parameters* | |
| Optimizer | AdamW |
| Learning Rate | $5 \times 10^{-5}$ (Full FT), $5 \times 10^{-4}$ (LoRA and FVAE-LoRA) |
| LR Scheduler | Linear |
| Warmup Steps | 500 |
| Batch Size | 32 |
| Number of Epochs | 30 |
| Weight Decay | 0.005 |
| CTC Loss Reduction | Sum |
| *LoRA Parameters (Standard LoRA & FVAE-LoRA's LoRA part)* | |
| LoRA Rank ($r$) | 16 |
| LoRA Dropout | 0.1 |
| *FVAE-LoRA Specific Parameters* | |
| Latent Dim. $\mathbf{z}_1$ | 16 |
| Latent Dim. $\mathbf{z}_2$ | 16 |
| FVAE $q_{\phi_i}(\mathbf{z}_i|\mathbf{x})$ Enc. Arch. | $\mathbf{x} \xrightarrow{\text{Linear}} \dim(\mathbf{z}_i) \xrightarrow{\text{ReLU}} \text{HiddenState}_{\mathbf{z}_i} \xrightarrow{\text{Linear}} (\boldsymbol{\mu}_{\mathbf{z}_i}, \log \boldsymbol{\sigma}_{\mathbf{z}_i}^2)$ |
| FVAE $p_\theta(\mathbf{x}|\mathbf{z}_1, \mathbf{z}_2)$ Dec. Arch. | $\text{Concat}(\mathbf{z}_1, \mathbf{z}_2) \xrightarrow{\text{Linear}} H_D = 128 \xrightarrow{\text{ReLU}} \text{Linear} \to \hat{\mathbf{x}}$ (Input Dim) |
| Prior $p_1(\mathbf{z}_1)$ | $\mathcal{N}(0, I)$ |
| Prior $p_2(\mathbf{z}_2)$ | $\mathcal{N}(1.5, I)$ |
| $\boldsymbol{\lambda}$ (Eq. 6) | $1 \times 10^{-3}$ |
| ELBO Coeff. $\alpha$ (Reconstr.) | 1 |
| ELBO Coeff. $\beta$ (KL $q_1||p_1$) | 1 |
| ELBO Coeff. $\delta$ | 1 |

## D.4 Spurious Correlation Experiments

These experiments (Waterbirds, CelebA, Animals) used ViT-B/16 as the backbone. Base training and LoRA parameters are similar to those in Section D.1, with specific FVAE-LoRA coefficients tuned for robustness.

Table 12: Key FVAE-LoRA Hyperparameters for Spurious Correlation tasks (ViT-B/16).

| Parameter | Value / Setting |
|---|---|
| *General & LoRA Parameters* | |
| See Table 8 for most general and LoRA parameters. | |
| Batch Size | 128 |
| Number of Epochs | 30 |
| *FVAE-LoRA Specific Parameters* | |
| Latent Dim. $\mathbf{z}_1$ | 16 |
| Latent Dim. $\mathbf{z}_2$ | 16 |
| FVAE Architecture | Similar to Table 8 |
| Prior $p_1(\mathbf{z}_1)$ | $\mathcal{N}(0, I)$ |
| Prior $p_2(\mathbf{z}_2)$ | $\mathcal{N}(1.5, I)$ |
| $\boldsymbol{\lambda}$ (Eq. 6) | $1 \times 10^{-3}$ or $1 \times 10^{-4}$ |
| ELBO Coeff. $\alpha$ (Reconstr.) | 0.1 or 1 |
| ELBO Coeff. $\beta$ (KL $q_1||p_1$) | 1 or 10 |
| ELBO Coeff. $\delta$ | 1 |

# E Early Attempts at Latent Space Factorization

The most straightforward way to enforce repulsion between $\mathbf{z}_1$ and $\mathbf{z}_2$ in the two-variable ELBO (see Eq. 2) would be to augment the ELBO with the terms:

$$+D_{\mathrm{KL}}\left(q_{\phi_2} \,\|\, p_1\right) + D_{\mathrm{KL}}\left(q_{\phi_1} \,\|\, p_2\right). \tag{9}$$

However, early experiments with this approach yielded poor results, in fact, performance was worse than with LoRA. From a theoretical standpoint, adding such terms to the two-variable ELBO effectively cancels out $q_{\phi_2}$ and $q_{\phi_1}$ from the objective, leading instead to a direct repulsion between the priors $p_1$ and $p_2$, which is not desirable. Other similar approaches such as directly repelling $q_{\phi_1}$ and $q_{\phi_2}$ suffered from the same issue. We found that all overly symmetric and direct formulations, including two-term symmetric variants, were ultimately unfruitful. To avoid this cancellation effect, we instead propose an indirect way to introduce repulsion between $\mathbf{z}_1$ and $\mathbf{z}_2$ by introducing a cross-term between the parametric encoder $q_{\phi_2}$ and the latent distribution $p_1$. This solution is theoretically grounded: we show that it induces a geometric separation, measured through a Wasserstein upper bound, between the two encoders $q_{\phi_1}$ and $q_{\phi_2}$. It is also supported by experimental results, outperforming LoRA across all tested modalities.

# F Additional Insights in FVAE-LoRA

Regarding the objective in Eq. 4, our proposed loss is a novel objective derived from and inspired by the Evidence Lower Bound, but it is not a strict lower bound on the marginal log-likelihood $\log p(\mathbf{x})$. By introducing the repulsive regularization term $\Gamma$, we modify the standard ELBO to enforce factorization between the latent spaces. This term is essential for the method's success, but it means the objective no longer serves as a formal lower bound on the data log-likelihood in the traditional VAE sense.

FVAE-LoRA intentionally sacrifices static weight merging to enable a more powerful dynamic, input-dependent adaptation. By computing the adaptation specifically for each input $\mathbf{x}$, our model learns more robust and fine-grained representations. We believe this dynamic mechanism is the key to its performance edge, a capability validated by our strong results on the spurious correlation benchmarks. This trade-off is therefore central to achieving the higher performance and robustness we demonstrate.

Note that simply reducing the rank of LoRA is a simple and efficient form of regularization, however It compresses all information flowing through the adapter, without distinguishing between features that are useful, irrelevant, or even detrimental to the downstream task. Our hypothesis is that large foundation models, pretrained on vast and general datasets, contain rich and entangled set of features. For any specific downstream task, some features are highly relevant (the "signal"), some are irrelevant but harmless, and some are actively harmful. The most prominent example of these detrimental features are spurious correlations (e.g., a water background being correlated with a "waterbird" label). A standard fine-tuning process, which optimizes a task-specific loss, may still latch onto these spurious features because they are prevalent in the training data and help minimize the training loss. This leads to poor generalization on data where that correlation is broken. This is why FVAE-LoRA is designed to be a more intelligent filter. Its goal is not just to compress, but to actively separate and isolate these different types of information. By using two latent spaces ($\mathbf{z}_1$ and $\mathbf{z}_2$) and our novel factorization objective, we encourage the model to encode task-salient, causal information in $\mathbf{z}_1$ while relegating the residual, non-essential, or spurious information to $\mathbf{z}_2$. The most direct validation of this rationale is in our spurious correlation experiments (Section 4.2). These results show that FVAE-LoRA is significantly more robust to misleading features than standard LoRA, confirming that it successfully learns to rely on the core features isolated within $\mathbf{z}_1$. This ability to "denoise" the adaptation is why it ultimately achieves better and more reliable performance.

# G A Practical Guide on Hyperparamters Selection

The factorization in FVAE-LoRA is governed by a subtle equilibrium between reconstruction and regularization, enforced by our ELBO objective. The key hyperparameters, $\beta$ and $\delta$, control this balance.

$\beta$ **and the Task-Salient Space ($\mathbf{z}_1$).** The $\beta$ parameter controls the KL divergence on $\mathbf{z}_1$, our task-salient latent space. Its role is critical, as it enforces a structured and efficient representation of the task-salient features. To understand its impact, we experimented with a wide range of values.

A significantly lower value, such as $\beta = 0.1$, led to a drastic degradation in performance across all tasks. This is because a near-zero $\beta$ effectively removes the KL divergence term, freeing the encoder for $\mathbf{z}_1$ to learn an unconstrained and arbitrarily complex representation. This removes the crucial pressure for the learned posterior $q_{\phi_1}(\mathbf{z}_1|\mathbf{x})$ to align with the prior $p_1(\mathbf{z}_1)$, leading to overfitting and a loss of generalization. This result is not merely a poor tuning choice; it is critical evidence that enforcing this prior alignment is essential for learning a robust and meaningful task-salient space.

Conversely, we explored a much higher value of $\beta = 100$. While this yielded marginal improvements over $\beta = 10$ on some specific tasks, the gains were not significant enough to justify such a strong constraint. An overly large $\beta$ can create an information bottleneck, punishing the model so heavily for deviating from the prior that it struggles to encode sufficient task-specific information in $\mathbf{z}_1$.

This evidence from both extremes reveals a necessary balance. The optimal values, which we found to be in the range of 1 to 10, are large enough to enforce a structured, regularized space but not so large as to prevent the learning of useful features.

$\delta$ **and Latent Space Separation.** The $\delta$ parameter controls the strength of our repulsive regularizer, $\Gamma$, which is the primary mechanism for enforcing factorization between the task-salient space ($\mathbf{z}_1$) and the residual space ($\mathbf{z}_2$). Our empirical results consistently show that $\delta = 1$ provides sufficient repulsive force to achieve this separation effectively, as was demonstrated in the spurious correlation experiments. We recommend $\delta = 1$ as a robust and generally optimal default.

For practical application, users can start with $\delta = 1$ and tune $\beta$ (typically between 1 and 10) to adjust the regularization on the learned task-salient features. This is a crucial point for the practical application of our method.

## H    Computational Cost Analysis

Our empirical results on image classification tasks show that the training time for FVAE-LoRA is approximately 30% higher than that of the strong DoRA baseline. This increase is primarily due to the additional forward and backward pass through the VAE's decoder during the training phase. However, the inference-time overhead is significantly lower because only the lightweight $\mathbf{z}_1$ encoder is used.

## I    Future Work

A particularly exciting avenue for future work lies in exploiting the inherent generative capabilities of the FVAE framework. Key directions will include exploiting the generative capabilities of the FVAE decoder for principled data augmentation, applying our latent factorization principle to other PEFT methods beyond LoRA, exploring approximate high-rank adaptation methods like HiRA, and exploring architectural enhancements such as allocating adaptive parameter budgets or different latent space ranks to different layers.

## J    Limitations

While FVAE-LoRA demonstrates promising results across diverse modalities, several limitations of the current work remain. First, the LoRA rank is fixed to a value of 16 across all experiments. Although this ensures consistent parameter budgets, it may not represent the optimal configuration for each task or domain, potentially limiting performance. Second, FVAE-LoRA and all baselines are applied only to the query and key matrices of the transformer models. This restricted application may overlook potential gains from adapting other components such as value matrices or feedforward layers.

Furthermore, detailed hyperparameter settings and VAE-specific architectural choices are provided in the appendix. This separation may hinder reproducibility, for readers interested in extending the approach. Nevertheless, the code will be open-sourced after publication.

In addition, while modality-specific baselines are chosen with the goal of providing meaningful comparisons, we do not evaluate against stronger non-LoRA or non-factorized alternatives, which may offer a more comprehensive picture of relative performance. Further, the paper does not report the computational cost of training or inference, which is important for assessing the practical deployment potential of the method, especially in resource-constrained environments.

Finally, a practical limitation of FVAE-LoRA is that its adapter weights cannot be merged back into the original base model after training. This stands in contrast to some LoRA-based methods that allow for such weight merging, which can simplify inference or reduce model complexity at deployment time.

## K   Broader Impacts

FVAE-LoRA advances parameter-efficient fine-tuning by enabling a factorized latent representation. Potential positive impacts include more effective and robust model adaptation across modalities, potentially leading to improved performance, resource efficiency, and more reliable AI systems, especially in handling spurious correlations, as demonstrated in our experiments. This can also enhance accessibility to powerful AI capabilities for a wider range of researchers and developers.

However, techniques that improve the adaptability of large foundation models also carry inherent risks. Easier and more effective fine-tuning could lower barriers for misuse in sensitive areas such as the generation of sophisticated disinformation or the development of enhanced surveillance tools. While FVAE-LoRA aims to disentangle task-salient information, it does not inherently mitigate biases that may be present in the original pre-trained models or the fine-tuning data. Indeed, such biases could potentially be concentrated or even amplified within the task-salient latent space if not proactively identified and addressed.

