# OpenReview forum: "Latent Space Factorization in LoRA"
_NeurIPS.cc/2025/Conference — NeurIPS 2025 poster_

### Official Review · Reviewer_aAk2 · 2025-06-19

**Clarity:** 2
**Significance:** 2
**Originality:** 3
**Rating:** 4
**Confidence:** 4

**Summary:**

This paper proposes FVAE-LoRA, which combines LoRA with a factorized VAE framework. The key idea is to learn two distinct latent spaces (z1 for task-salient information and z2 for residual information) through a novel ELBO, with only z1 being used for downstream task adaptation. The FVAE-LoRA is extensively evaluated across diverse tasks.

**Questions:**

1. In Equation (3), why does the regularization term specifically penalize the similarity between qφ2(z2|x) and p1(z1) rather than other possible combinations? Since both p1 and p2 are normal distributions, what is the specific theoretical or empirical rationale for encouraging qφ2 to be dissimilar from p1 specifically?

2. Why is p2 specifically centered at 1.5? The paper provides no theoretical justification or empirical analysis for this choice. Was this value determined through hyperparameter search, theoretical considerations, or some other principle?

**Ethical Concerns:**

["NO or VERY MINOR ethics concerns only"]

**Final Justification:**

The authors' rebuttal adequately addresses the efficiency concerns and demonstrates that the approach achieves meaningful improvements on challenging tasks requiring robust feature separation.

**Limitations:**

yes

**Paper Formatting Concerns:**

N/A.

**Quality:**

2

**Strengths And Weaknesses:**

Strengths:

1. The integration of factorized VAE with LoRA is conceptually interesting, and the formulation with the cross-prior regularization term (Γ) is well-motivated.

2. The experimental evaluation spans multiple modalities (vision, text, audio) and includes diverse datasets, demonstrating the generality of the approach.

3. The spurious correlation experiments provide compelling evidence that the method indeed learns more robust representations.

Weaknesses:

1. FVAE-LoRA replaces LoRA's simple linear projections with complex multi-layer nonlinear VAE mappings, and uses two separate VAEs, substantially increasing computational complexity. While the authors acknowledge this in the limitations, they provide no analysis of the actual computational costs or training time overhead. For a PEFT method, this lack of efficiency analysis is a critical omission.

2. The non-mergeable weights represent a severe practical limitation that undermines one of LoRA's core advantages. The ability to merge adapter weights back into the base model is a defining feature of reparameterization-based PEFT methods, enabling efficient deployment without inference overhead. By making the weights non-mergeable, FVAE-LoRA essentially transforms LoRA into an adapter-like method, losing this key benefit. Given the marginal performance improvements (e.g., 87.82% vs 87.40% on commonsense reasoning), these gains do not justify such a significant architectural limitation.

3. The method introduces multiple hyperparameters (α, β, δ, λ). The paper lacks systematic analysis of sensitivity to these choices.

4. Missing comparisons with non-LoRA PEFT methods like FourierFT.

---

> ### Author Response · Authors · 2025-08-01
> **Authors Response to Reviewer aAk2 (1/n)**
>
> We appreciate the reviewer’s valuable feedback and insightful comments. Below, we address each of the review points in detail.
>
> ### Weaknesses
>
> 1. While our method introduces a VAE framework for LoRA, its added complexity is carefully managed, particularly at inference time. To be precise, only the standard LoRA matrix $A$ is replaced by the encoder for the task-salient latent space, $z_1$. As detailed in the appendix, this encoder is a simple network consisting of just two linear layers with a non-linearity in between. The decoder and the entire pathway related to the residual latent space, $z_2$, are utilized only during training to enforce the factorization and are completely discarded at inference. This design ensures that the total number of trainable parameters remains comparable to standard LoRA, with the goal of using a similar parameter budget more effectively for superior performance.
>
> Regarding the practical overhead, our empirical results on image classification tasks show that the training time for FVAE-LoRA is approximately 30% higher than that of the strong DoRA baseline. This increase is primarily due to the additional forward and backward pass through the VAE's decoder during the training phase. However, the inference-time overhead is significantly lower because only the lightweight $z_1$ encoder is used.
>
> 2. The non-mergeable weights in FVAE-LoRA were a deliberate trade-off introduced to enable a richer representation of the latent space via the VAE framework. While this design choice may limit the ability to merge weights, it opens up opportunities for more nuanced adaptation and generalization strategies.
>
> The average performance on the Commonsense Reasoning task (87.82% for FVAE vs. 87.40% for HIRA) should not be viewed as the sole criterion for assessing the value of our method. FVAE-LoRA introduces a more expressive architecture with potential advantages beyond this specific benchmark. It demonstrates a clear benefit on more complex tasks, where richer, non-linear representations lead to more substantial improvements. For instance, we observe notable gains on challenging datasets such as SUN397 (complex scenes), WNLI (coreference resolution), and the Animals spurious correlation benchmark. These results highlight that, while standard LoRA performs adequately on simpler tasks, our explicit factorization offers a key advantage in scenarios where distinguishing the true task signal from noise is especially difficult—resulting in a more robust and capable model overall.
> That said, we recognize that the non-mergeable weights could impact deployment in resource-constrained environments. Addressing this limitation is an important direction for future work, potentially through hybrid approaches that allow for some degree of parameter merging post-training.
>
> To provide a complete picture of the efficiency trade-offs, we will add a table to the appendix comparing the wall-clock training/inference times of FVAE-LoRA with baselines for completeness.

---

> > ### Author Response · Authors · 2025-08-01
> > **Authors Response to Reviewer aAk2 (2/n)**
> >
> > 3. The factorization in FVAE-LoRA is governed by a subtle equilibrium between reconstruction and regularization, enforced by our ELBO objective. The key hyperparameters, β and δ, control this balance.
> >
> > **β (Beta) and the Task-Salient Space ($z_1$)**: The β parameter controls the KL divergence on $z_1$, our task-salient latent space. Its role is critical, as it enforces a structured and efficient representation of the task-salient features. To understand its impact, we experimented with a wide range of values.
> >
> > A significantly lower value, such as **β = 0.1**, led to a **drastic degradation in performance across** all tasks. This is because a near-zero β effectively removes the KL divergence term, freeing the encoder for $z_1$ to learn an unconstrained and arbitrarily complex representation. This removes the crucial pressure for the learned posterior $q(z_1|x)$ to align with the prior $p(z_1)$,, leading to overfitting and a loss of generalization. This result is not merely a poor tuning choice; it is critical evidence that **enforcing this prior alignment is essential** for learning a robust and meaningful task-salient space.
> >
> > Conversely, we explored a much higher value of **β = 100**. While this yielded marginal improvements over **β = 10** on some specific tasks, the gains were not significant enough to justify such a strong constraint. An overly large β can create an information bottleneck, punishing the model so heavily for deviating from the prior that it struggles to encode sufficient task-specific information in $z_1$.
> >
> > This evidence from both extremes reveals a necessary balance. The optimal values, which we found to be in the **range of 1 to 10**, are large enough to enforce a structured, regularized space but not so large as to prevent the learning of useful features.
> >
> > **δ (Delta) and Latent Space Separation**: The δ parameter controls the strength of our repulsive regularizer, Γ, which is the primary mechanism for enforcing factorization between the task-salient space ($z_1$) and the residual space ($z_2$). Our empirical results consistently show that **δ = 1** provides sufficient repulsive force to achieve this separation effectively, as was demonstrated in the spurious correlation experiments. We recommend **δ = 1** as a robust and generally optimal default.
> >
> > For practical application, users can start with **δ = 1** and tune **β** (typically between 1 and 10) to adjust the regularization on the learned task-salient features.
> > This is a crucial point for the practical application of our method. We will add a dedicated subsection in the appendix to provide a comprehensive guide on hyperparameter selection.
> >
> > 4. Our primary focus was to directly build upon the LoRA framework, addressing what we identified as a lacunae in its update mechanism. Consequently,  to the best of our knowledge, we prioritized a rigorous comparison against the strongest LoRA variations (like DoRA and HiRA) across multiple modalities to clearly validate our core contribution.
> >
> > In the context of other methods like FourierFT, our reading of the original work indicates its performance on text benchmarks is generally comparable to standard LoRA. Since FVAE-LoRA consistently and significantly outperforms standard LoRA, this already establishes a strong benchmark for its performance in relation to FourierFT. We believe our comprehensive comparisons against the most relevant and powerful baselines within the LoRA family effectively situate our method's advancements.
> >
> >
> > ### Questions
> > 1. The most straightforward way to enforce repulsion between $z_1$ and $z_2$ in the two-variable ELBO (see Eq. 2) would be to augment the ELBO with the terms:
> >
> > \+ DKL($q_{\phi_2} || p_1$) + DKL($q_{\phi_1} || p_2$).
> >
> > However, early experiments with this approach yielded poor results, in fact, performance was worse than with LoRA. From a theoretical standpoint, adding such terms to the two-variable ELBO effectively cancels out $q_{\phi_2}$ and $q_{\phi_1}$ from the objective, leading instead to a direct repulsion between the priors$ p(z_1)$ and $p(z_2)$, which is not desirable.
> > Other similar approaches such as directly repelling $q_{\phi_1}$ and $q_{\phi_2}$  suffered from the same issue. We found that all overly symmetric and direct formulations, including two-term symmetric variants, were ultimately unfruitful.
> > To avoid this cancellation effect, we instead propose an indirect way to introduce repulsion between $z_1$ and $z_2$ by introducing a cross-term between the parametric encoder $q_{\phi_2}$ and the latent distribution $p_1$. This solution is theoretically grounded: we show that it induces a geometric separation, measured through a Wasserstein upper bound, between the two encoders $q_{\phi_1}$ and $q_{\phi_2}$. It is also supported by experimental results, outperforming LoRA across all tested modalities.
> >
> > We will include an appendix detailing our early attempts to better understand and justify the choices and derivations in the paper.

---

> ### Author Response · Authors · 2025-08-01
> **Authors Response to Reviewer aAk2 (n/n)**
>
> 2. The intuition is to give the two priors distinct non-overlapping "location" in the latent space to initialize and encourage separation. By setting $\mu_1$₁ at 0 and $\mu_2$ at 1.5, we provide a clear signal for the repulsive regularizer to push the posteriors apart. We agree that a more in-depth empirical study of prior initializations would be a valuable piece of future work to better understand and optimize this factorization.
>
> We hope we have addressed the reviewer's concerns and would be happy to provide any additional details if needed. We sincerely hope the reviewer will consider revising the scores in light of the clarifications provided.

---

> > ### Comment · Reviewer_aAk2 · 2025-08-05
> >
> > I thank the authors for the comprehensive rebuttal, which has largely addressed my concerns. The clarification about inference-time complexity (only using the z1 encoder) and the 30% training overhead helps with the efficiency concern. The hyperparameter sensitivity analysis for β and δ is also helpful.

---

### Official Review · Reviewer_PQZk · 2025-06-21

**Clarity:** 3
**Significance:** 2
**Originality:** 2
**Rating:** 4
**Confidence:** 3

**Summary:**

This paper introduces FVAE-LoRA, a new method to improve LoRA, by using a VAE to learn two separate latent spaces. FVAE-LoRA explicitly separate task-relevant from irrelevant residual features. They design a novel ELBO to encourage a clean separation between these spaces. They demonstrate the effectiveness of FVAE-LoRA by performing extensive experiments across text, audio, and vision, especially in scenarios involving spurious correlations or distribution shifts.

**Questions:**

1. What are the recommended ranges/values of these hyperparameters (e.g. \beta, \delta)? Would be great to see a guide on how to choose these hyperparameters. How sensitive are the empirical results to different hyperparameter choices?

2. Could you add some visualization of what is actually encoded in z1 vs. z2? I'm wondering if visualization shows z1 captures task-related information, while z2 captures residual as expected?

3. Could you add some training/inference time comparisons between the proposed method and existing methods?

**Ethical Concerns:**

["NO or VERY MINOR ethics concerns only"]

**Final Justification:**

I thank the authors for their detailed responses to my questions and effort to add more useful results and discussions in appendix. I'm willing to increase my score to 4.

**Limitations:**

Could you add some future work discussions in the paper?

**Paper Formatting Concerns:**

none.

**Quality:**

2

**Strengths And Weaknesses:**

Strengths:
- the paper introduces a carefully designed objective and has run extensive empirical experiments to demonstrate the effectiveness the proposed method.
- the paper is clearly written and easy to follow.

Weakness:
- the improvements of proposed method compared to LoRA are small on some benchmarks, sometimes the improvements are within the error bars range (e.g. WATERBIRDS).
- error bars (stddev) are missing in Table 2.
- lack of hyperparams tuning guide and analysis.
- lack of interpretability.
- lacks computation complexity analysis. it's unclear how much training time is increased compared to LoRA.

---

> ### Author Response · Authors · 2025-08-01
> **Authors Response to Reviewer PQzk**
>
> We appreciate the reviewer’s valuable feedback and insightful comments. Below, we address each of the review points in detail.
>
> - **Weakness 1** *the improvements of proposed method compared to LoRA are small on some benchmarks, sometimes the improvements are within the error bars range (e.g. WATERBIRDS).*
>
> While the performance gain on certain individual datasets, such as Waterbirds for Worst-Group Accuracy (Table 5), are within error bars, FVAE-LoRA establishes a clear and consistent pattern of positive performance across all evaluated tasks and modalities.
>
> The central hypothesis of our work is that explicitly factorizing the low-rank update to isolate task-salient information leads to a more effective adaptation. The results strongly validate this hypothesis. We see its benefits manifest in two crucial ways. First, FVAE-LoRA consistently achieves a superior **average** performance across entire benchmarks, including the comprehensive image classification suite (Table 1) and the GLUE benchmark (Table 3). This demonstrates the overall reliability and effectiveness of our more principled update mechanism.
>
> Second, and perhaps more importantly, the advantage of our method becomes most pronounced on the most challenging tasks. We observe significant performance improvements on difficult datasets like SUN397 (complex scenes), WNLI (coreference resolution), and the Animals spurious correlation benchmark. This shows that while standard LoRA may perform adequately on simpler tasks, our explicit factorization provides a critical advantage where distinguishing the true task signal from noise is most difficult, resulting in a more robust and capable model overall.
>
> - **Weakness 2** *error bars (stddev) are missing in Table 2.*
>
> For the Llama-3-8B experiments, we followed the evaluation protocol of prior work, such as the HiRA paper, due to the high computational cost of multiple fine-tuning runs. To ensure a direct and fair comparison, we used the same single seed reported in HiRA paper, a common practice for evaluations on models of this scale and size.
>
>  - **Other Weakness** Answered in the questions section.

---

> ### Author Response · Authors · 2025-08-01
> **Authors Response 2 to Reviewer PQzk**
>
> - **[Q1]** *What are the recommended ranges/values of these hyperparameters (e.g. \beta, \delta)? Would be great to see a guide on how to choose these hyperparameters. How sensitive are the empirical results to different hyperparameter choices?*
>
> The factorization in FVAE-LoRA is governed by a subtle equilibrium between reconstruction and regularization, enforced by our ELBO objective. The key hyperparameters, β and δ, control this balance.
>
> **β (Beta) and the Task-Salient Space ($z_1$)**: The β parameter controls the KL divergence on $z_1$, our task-salient latent space. Its role is critical, as it enforces a structured and efficient representation of the task-salient features. To understand its impact, we experimented with a wide range of values.
>
> A significantly lower value, such as **β = 0.1**, led to a **drastic degradation in performance across** all tasks. This is because a near-zero β effectively removes the KL divergence term, freeing the encoder for $z_1$ to learn an unconstrained and arbitrarily complex representation. This removes the crucial pressure for the learned posterior $q(z_1|x)$ to align with the prior $p(z_1)$, leading to overfitting and a loss of generalization. This result is not merely a poor tuning choice; it is critical evidence that **enforcing this prior alignment is essential** for learning a robust and meaningful task-salient space.
>
> Conversely, we explored a much higher value of **β = 100**. While this yielded marginal improvements over **β = 10** on some specific tasks, the gains were not significant enough to justify such a strong constraint. An overly large β can create an information bottleneck, punishing the model so heavily for deviating from the prior that it struggles to encode sufficient task-specific information in $z_1$.
>
> This evidence from both extremes reveals a necessary balance. The optimal values, which we found to be in the **range of 1 to 10**, are large enough to enforce a structured, regularized space but not so large as to prevent the learning of useful features.
>
> **δ (Delta) and Latent Space Separation**: The δ parameter controls the strength of our repulsive regularizer, Γ, which is the primary mechanism for enforcing factorization between the task-salient space ($z_1$) and the residual space ($z_2$). Our empirical results consistently show that **δ = 1** provides sufficient repulsive force to achieve this separation effectively, as was demonstrated in the spurious correlation experiments. We recommend **δ = 1** as a robust and generally optimal default.
>
> For practical application, users can start with **δ = 1** and tune **β** (typically between 1 and 10) to adjust the regularization on the learned task-salient features.
> This is a crucial point for the practical application of our method. We will add a dedicated subsection in the appendix to provide a comprehensive guide on hyperparameter selection.
>
> - **[Q2]** *Could you add some visualization of what is actually encoded in z1 vs. z2? I'm wondering if visualization shows z1 captures task-related information, while z2 captures residual as expected?*
>
> Thank you for this suggestion. A new visualization section will be added to the appendix. This section will present reconstructions generated separately from the task-salient latent space ($z_1$) and the residual latent space ($z_2$).
>
> - **[Q3]** *Could you add some training/inference time comparisons between the proposed method and existing methods?*
>
> We thank the reviewer for this suggestion. A table comparing the empirical training and inference time measurements for FVAE-LoRA against the relevant baselines will be added to the appendix.
>
> - **Limitations** *Could you add some future work discussions in the paper?*
>
> A particularly exciting avenue for future work lies in exploiting the inherent **generative capabilities** of the FVAE framework. Key directions will include exploiting the generative capabilities of the FVAE decoder for principled data augmentation, applying our latent factorization principle to other PEFT methods beyond LoRA, exploring approximate high-rank adaptation methods like HiRA, and exploring architectural enhancements such as allocating adaptive parameter budgets or different latent space ranks to different layers. A more detailed discussion on these points will be added to the appendix.
>
> We hope we have addressed the reviewer's concerns and would be happy to provide any additional details if needed. We sincerely hope the reviewer will consider revising the scores in light of the clarifications provided.

---

> > ### Author Response · Authors · 2025-08-07
> > **Authors Response to Reviewer PQzk**
> >
> > Dear Reviewer PQzk,
> >
> > As the discussion period concludes tomorrow, we wanted to kindly follow up to see if our rebuttal has addressed your concerns. If there are any remaining questions or if further clarification would be helpful, we would be more than happy to provide it.
> >
> > Thank you again for your time and feedback.
> >
> > Best regards,
> >
> > The Authors

---

> > ### Comment · Reviewer_PQZk · 2025-08-08
> > **Response to author rebuttal**
> >
> > I thank the authors for their detailed responses to my questions and effort to add more useful results and discussions in appendix. I'm willing to increase my score to 4.

---

### Official Review · Reviewer_5sDq · 2025-06-30

**Clarity:** 4
**Significance:** 3
**Originality:** 3
**Rating:** 5
**Confidence:** 4

**Summary:**

In this papers the authors propose a novel PEFT technique based on the popular LoRA method (low rank adaptation), that leverages latent disentanglement obtained via variational auto-encoders (VAE) and a novel disentanglement loss (FVAE-LoRA). In LoRA, one fine-tunes a low rank decomposition $\mathbf{B} \mathbf{A}$ ($\mathbf{A} \in \mathbb{R}^{r \times d}$, \mathbf{B} \in \mathbb{R}^{k \times r}, with the rank $r << \min(k, d)$) of linear weights that is summed to the frozen weights of the base model. The authors suggest instead to replace the projection $\mathbf{A}$ with two joint VAE posteriors $q\_1$, $q\_2$ mapping to two latent variables $\mathbf{z}\_1$, $\mathbf{z}\_2$, that reconstruct the activation via a decoder $p_{\theta}$. The $\mathbf{B}$ matrix only maps back in the architecture the $\mathbf{z}\_1$ architecture, which is disentangled from $\mathbf{z}\_2$ via a novel ELBO loss, which penalises the posterior $q\_2$ to match the prior of $\mathbf{z}_1$ ($p_1$), by maximising their KL divergence (Eq. 3). This formulation contains a  $\Gamma$ modulator term, that can be interpreted as composed of two parts, *mismatch* and *discrepancy*, with the later being a lower bound on the 2-Wasserstein distance between the two posteriors. The authors conduct a thorough empirical study, showcasing strong performance and robustness to spurious correlations.

**Questions:**

- What I don’t really understand is why factoring your latent space into two Gaussians would lead to better performance. Couldn’t the same be achieved by simply reducing the rank? If not, I understand that the rationale is that if the $\mathbf{z}_1$ features live on a manifold, this could lead to better representations for the downstream task.  Maybe the authors could touch more on this rationale, better interpreting their proposed method.
- Related to this, the authors say in the introduction (Line 25) that it is not guaranteed that the low-rank projections fully capture task related tasks. Why should those features be detrimental if one optimises a loss specifically for the task during fine-tuning?
- Did the authors tried using more than two auto-encoders for the factorisation?
- Do the authors think that a more complex posterior than Gaussian would improve performance?
- Line 99: I don’t think one has to put into the appendix the derivation of the ELBO since it is pretty standard knowledge in deep learning literature (e.g., VAE paper).
- Eq. 5: Should we have $\mathbf{z}_2$ instead of $\mathbf{z}$?
- Why this choice for centering $p_2$ on $\mathbf{1.5}$? Dose it encourages factorization? It would have been interesting if the authors studied more in depth the prior initialisations.
- In Table 6, the VAE2LAT loss is referenced in Eq. 2, not 3.

**Ethical Concerns:**

["NO or VERY MINOR ethics concerns only"]

**Final Justification:**

The authors proposed a nove PEFT technique based on latent space disentanglement. The answer of the authors provided more cues into why such a method showcases good empirical results.

**Limitations:**

yes

**Quality:**

3

**Strengths And Weaknesses:**

## Strengths

### Novelty
As of my knowledge, doing disentanglement in a PEFT method is a novel idea, so I appreciate the contribution. At the same time it seems very surprising that such a method should work at all since the variational formulation, while useful for factoring the low rank space, it does not seem directly related to the task at hand (see first two Questions). I can see it justified by the fact that when using bot the task loss and the VAE loss, only $\mathbf{z}\_1$ is involved into computing the task loss, so this should compress more task-related information into it. But the same could be applied directly for example by simply lowering the rank (so as to not store spurious information into the low-rank space). Nevertheless, the good empirical results suggest that such a method really benefits from the variational loss. This method could as well be extended for example into optimisation methods that leverage singular value decomposition for reducing the size of a network, in order to better capture useful representations during miniaturization.

### Soundness
The methodology is mathematically sound and well posed. I like the decomposition into mismatch and discrepancy, and the proof which shows that maximising the discrepancy maximises a geometric Wasserstein disstance between the two posteriors.

### Strong experimental evidence
The method is tested across a plethora of domains (vision, text, audio) and showcases improved performance with respect to state-of-the-art methods in efficient fine tuning.

## Weaknesses

### Interpretation
As noted in the previous part, the authors should better interpret why disentanglement is key into achieving better performance as showcased by their empirical studies. Also it is not clear
if the proposed loss (Eq. 4) is truly an ELBO as the author claim (is it still a lower bound of the log likelihood?).

### Computational / complexity
As the authors suggest into the limitation section of their work, one key aspect of LoRA, being able to merge the fine-tuned weights with the original frozen weights by simply summing
them, it is not possible with the current method. This could hinder the applicability of such a method, since it requires separate layers for the adapters.

---

> ### Author Response · Authors · 2025-08-01
> **Authors Response to Reviewer 5sDq (1/2)**
>
> We sincerely thank the reviewer for their thorough and constructive feedback.
>
> ### Weaknesses
>
> - **Interpretation**: We agree that a clear interpretation of why our method works is crucial. The core rationale is addressed in detail in our answer to your first question below.
>
> Regarding the objective in Eq. 4, our proposed loss is a novel objective **derived from and inspired by the Evidence Lower Bound**, but it is not a strict lower bound on the marginal log-likelihood log p(x). By introducing the repulsive regularization term $\gamma$, we modify the standard ELBO to enforce factorization between the latent spaces. This term is essential for the method's success, but it means the objective no longer serves as a formal lower bound on the data log-likelihood in the traditional VAE sense. We will ensure this distinction is made clear in the revised version of the paper.
>
> - **Computational / complexity**: We acknowledge this trade-off. FVAE-LoRA intentionally sacrifices static weight merging to enable a more powerful **dynamic, input-dependent adaptation**.
> By computing the adaptation specifically for each input x, our model learns more robust and fine-grained representations. We believe this dynamic mechanism is the key to its performance edge, a capability validated by our strong results on the spurious correlation benchmarks. This trade-off is therefore central to achieving the higher performance and robustness we demonstrate.
>
> ### Questions
>
> - **[Q1, Q2]**: Simply reducing the rank is a simple and efficient form of regularization, however It compresses all information flowing through the adapter, without distinguishing between features that are useful, irrelevant, or even detrimental to the downstream task.
>
> Our core hypothesis is that large foundation models, pretrained on vast and general datasets, contain rich and entangled set of features. For any specific downstream task, some features are highly relevant (the "signal"), some are irrelevant but harmless, and some are **actively harmful**. The most prominent example of these detrimental features are **spurious correlations** (e.g., a water background being correlated with a "waterbird" label). A standard fine-tuning process, which optimizes a task-specific loss, may still latch onto these spurious features because they are prevalent in the training data and help minimize the training loss. This leads to poor generalization on data where that correlation is broken.
>
> This is why FVAE-LoRA is designed to be a more intelligent filter. Its goal is not just to compress, but to **actively separate and isolate** these different types of information. By using two latent spaces ($z_1$ and $z_2$) and our novel factorization objective, we encourage the model to encode task-salient, causal information in $z_1$ while relegating the residual, non-essential, or spurious information to $z_2$.
>
> The most direct validation of this rationale is in our **spurious correlation experiments (Section 4.4)**. These results show that FVAE-LoRA is significantly more robust to misleading features than standard LoRA, confirming that it successfully learns to rely on the core features isolated within $z_1$. This ability to "denoise" the adaptation is why it ultimately achieves better and more reliable performance.
>
> - **[Q3]**: Using the VAE2LAT model (equation 2) is equivalent to using two autoencoders with a common decoder (using the two latent variables). As shown in Table 6, this model does not yield good results, the lack of a repulsive term performs significantly worse than both our proposed FVAE-LoRA and even standard LoRA. This demonstrates that simply adding latent spaces is insufficient; it is the specific objective that encourages meaningful factorization that drives the performance gains. We focused on the fundamental two-space factorization (task-salient vs. residual) as it is the cleanest and most direct way to validate our core hypothesis.
>
> - **[Q4]**: We believe that employing more expressive posteriors (e.g., Normalizing Flows, etc.), has the potential to capture more complex latent structures and could further improve performance. However, for this work, and to be in line with the spirit of the LoRA and VAE, we decided to choose a simple diagonal Gaussian posterior for two key reasons: **ensure a fair and direct comparison** by keeping the architecture simple, and **to maintain computational efficiency expected from a LoRA model**.
>
> - **[Q5]**: We included it for the sake of completeness, but we agree it is standard knowledge and could be omitted for brevity.
>
> - **[Q6]**: Thank you for catching this typo, which we will correct.

---

> > ### Author Response · Authors · 2025-08-01
> > **Authors Response to Reviewer 5sDq (2/2)**
> >
> > - **[Q7]**: The intuition is to give the two priors distinct non-overlapping "location" in the latent space to initialize and encourage separation. By setting $\mu_1$₁ at 0 and $\mu_2$ at 1.5, we provide a clear signal for the repulsive regularizer to push the posteriors apart. We agree that a more in-depth study of prior initializations would be a valuable piece of future work to better understand and optimize this factorization.
> >
> > - **[Q8]** Thank you. We will correct this typo.

---

> > > ### Comment · Reviewer_5sDq · 2025-08-07
> > >
> > > I appreciate the additional insights over the interpretation of the method, that should flow in the final version of the paper. While the method still seems a little bit difficult to interpret, it's good result showcase that it is a technique worthy of publication / future investigation. As such I keep my full accept score.

---

### Official Review · Reviewer_9e3a · 2025-07-02

**Clarity:** 3
**Significance:** 3
**Originality:** 3
**Rating:** 4
**Confidence:** 4

**Summary:**

This paper introduces FVAE-LoRA , a variant of LoRA that incorporates VAE components to learn two distinct latent spaces within each LoRA module. In FVAE-LoRA, the down-projection matrix $A$ is decomposed into two submatrices, which are then learned in a VAE-style framework. During inference, these two matrices can be merged into a single matrix. The method is evaluated across a range of image, text, and audio benchmarks to comprehensively assess its performance.

**Questions:**

I think overall it's a nice paper, though I have concerns to be addressed, see Weaknesses. My initial rating is BA.

**Ethical Concerns:**

["NO or VERY MINOR ethics concerns only"]

**Final Justification:**

The rebuttal addressed my concerns. Since no new evidence/results are reported, having read other reviews, I'd keep my original positive score of 4.

**Limitations:**

yes

**Quality:**

3

**Strengths And Weaknesses:**

Strengths:
* The paper is well-written and clearly structured.
* The idea of integrating a VAE into LoRA is interesting, VAE naturally brings more variabilities into the low-dimensional space, and the same inference format is an extra plus.
* The evaluation across diverse modalities, image, text, and audio tasks, demonstrates the versatility and general applicability of the method.

Weaknesses:
* Performance wise,  the advantage of FVAE-LoRA over HiRA is not clearly demonstrated. While both methods aim to factorize LoRA matrices (HiRA via a Hadamard product of $BA$ and $W_0$), HiRA appears to be simpler to train due to the absence of additional VAE-related loss terms. As a result, it remains unclear what practical benefits FVAE-LoRA offers over HiRA in standard evaluations.
* The integration of VAE into LoRA opens up promising directions beyond conventional fine-tuning tasks. It would be valuable to explore the broader capabilities of FVAE-LoRA in more specialized settings. For instance, the structured latent space introduced by the VAE could be particularly beneficial in continual learning scenarios, where maintaining separation between task-specific representations may help mitigate catastrophic forgetting. Similarly, the generative nature of the VAE could enable more interpretable and controllable manipulation of the low-rank space, which might be advantageous in tasks involving conditional generation etc. Personally, I think FVAE-LoRA might shine more on these kinds of tasks, rather than the standard fine-tuning benchmarks.

---

> ### Author Response · Authors · 2025-08-01
> **Authors Response to Reviewer 9e3a**
>
> We appreciate the reviewer’s valuable feedback and insightful comments. Below, we address each of the review points in detail.
>
> - **Weakness 1** *Performance wise, the advantage of FVAE-LoRA over HiRA is not clearly demonstrated. While both methods aim to factorize LoRA matrices (HiRA via a Hadamard product of  and ), HiRA appears to be simpler to train due to the absence of additional VAE-related loss terms. As a result, it remains unclear what practical benefits FVAE-LoRA offers over HiRA in standard evaluations.*
>
> Firstly, please note that HiRA does not perform matrix factorization in the manner of LoRA; rather, it leverages high-rank adaptation via Hadamard product. While the average accuracy improvement of FVAE-LoRA over HiRA may be modest in certain settings, the method offers several practical advantages. These stem from its factorization mechanism, which contributes to improved robustness and broad cross-modality effectiveness.
>
> **Different Mechanisms**: HiRA does not explicitly factorize the low rank matrices from LoRA, whereas FVAE-LoRA introduces a novel ELBO objective to learn a factorization. Our method is explicitly designed to compel the model to disentangle task-salient features $z_1$ from residual ones $z_2$.
>
> **The Practical Benefit of Robustness**: One primary advantage of this principled disentanglement is demonstrated in **Section 4.4 and Table 7**, where we evaluate the models on spurious correlation benchmarks. This focus on empirically-validated robustness is a key differentiator. While high-rank updates like HiRA's have proven effective on standard benchmarks, **there is no inherent mechanism or experimental validation to ensure that the learned updates are robust to spurious correlations**. In contrast, our semantic factorization is specifically designed for this purpose. The experiments in Section 4.4 provide this crucial validation, showing that FVAE-LoRA consistently outperforms standard LoRA by achieving significantly higher Worst-Group (WG) accuracy. For example, on the ANIMALS dataset, FVAE-LoRA improves WG accuracy by **7.21%** and on CelebA by **3.33%**. This provides a more reliable adaptation for challenging, real-world scenarios.
>
> **Broad Applicability Across Modalities**: Furthermore, our paper demonstrates that FVAE-LoRA consistently improves performance not just in one domain, but across diverse **image, text, and audio tasks** (as shown in Tables 2, 4, 5, and 6). This versatility is a significant practical advantage, as many PEFT variants are often evaluated or optimized for a single modality.
>
> Therefore, we argue that the key contribution and practical benefit of FVAE-LoRA is not just improvements in average performance, but its demonstrated value as a **versatile and robust PEFT framework**. Its unique factorization objective leads to more reliable models that are less susceptible to learning misleading shortcuts, while its effectiveness is validated across a wide range of modalities, addressing a common limitation of more specialized methods.
>
> - **Weakness 2** *The integration of VAE into LoRA opens up promising directions beyond conventional fine-tuning tasks. It would be valuable to explore the broader capabilities of FVAE-LoRA in more specialized settings. For instance, the structured latent space introduced by the VAE could be particularly beneficial in continual learning scenarios, where maintaining separation between task-specific representations may help mitigate catastrophic forgetting. Similarly, the generative nature of the VAE could enable more interpretable and controllable manipulation of the low-rank space, which might be advantageous in tasks involving conditional generation etc. Personally, I think FVAE-LoRA might shine more on these kinds of tasks, rather than the standard fine-tuning benchmarks.*
>
> We completely agree that FVAE-LoRA extends far beyond standard fine-tuning, and this framework may be a natural fit for challenges like **continual learning** and **controllable generation**.
>
> For **continual learning**, the explicit separation of a task-salient latent space $z_1$ could indeed be a powerful mechanism to mitigate catastrophic forgetting by isolating task-specific adaptations.
>
> For **controllable tasks**, the generative nature of the VAE offers a principled way to manipulate the low-rank space, potentially enabling more interpretable and steerable model behavior by operating on $z$.
>
> For this work, our goal was to first establish the core effectiveness and, crucially, the robustness of FVAE-LoRA on a wide range of standard benchmarks across multiple modalities. Having validated the method's benefits, we consider these as promising future directions.
>
> We hope we have addressed the reviewer's concerns and would be happy to provide any additional details if needed. We sincerely hope the reviewer will consider revising the scores in light of the clarifications provided.

---

> > ### Comment · Reviewer_9e3a · 2025-08-05
> >
> > Thanks for the response. In general I'm satisfied with the additional explanation. I'd keep my original positive assessment of this work.

---

### Decision · Program_Chairs · 2025-09-17

**Decision:**

Accept (poster)

**Comment:**

This paper claims that part of the gap in performance between LoRA and full fine-tuning may be attributable to the fact that LoRA does not explicitly promote its latent spaces to reflect task-specific information. To remedy this, the paper proposes to adjoin the LoRA component, at training time, with a secondary latent representation, design a loss function that encourages this secondary latent space to capture the residual information, such that both latent spaces together can reconstruct inputs. The hope is that the LoRA side, which is the only one used at inference-time, then specializes more acutely to task-specific information. The paper demonstrates widespread empirical advantage to this approach and also empirically validates that this kind of separation is indeed happening, supporting the argument that the advantage is thanks to it.

Reviewers generally appreciate the paper's insight, results, and clear expositions. Several added clarifications offered during the discussions should become part of the paper. The theoretical underpinnings of the method, both in terms of the proposed ELBO-like loss, whether it promotes what it claims, and why this indeed improves the overall optimization, remain a weak point. Indeed, the model space is unchanged compared to LoRA, and it would be very interesting to have some analytical hints as to why/when regularizing the optimization with a form of a multi-task loss, which has components of both reconstruction (of the input) and differentiation (of the two latent spaces), can help the primary task.